# Peripheral thickening of the sarcomeres and pointed end elongation of the thin filaments are both promoted by SALS and its formin interaction partners

**Dávid Farkas**[1], **Szilárd Szikora**[1], **A. S. Jijumon**[1¤], **Tamás F. Polgár**[2,3], **Roland Patai**[2], **Mónika Ágnes Tóth**[4], **Beáta Bugyi**[4], **Tamás Gajdos**[5], **Péter Bíró**[5], **Tibor Novák**[5], **Miklós Erdélyi**[5], **József Mihály**[1,6]*

1 Institute of Genetics, Biological Research Centre, Szeged, Hungary, 2 Institute of Biophysics, Biological Research Centre, Szeged, Hungary, 3 Doctoral School of Theoretical Medicine, University of Szeged, Szeged, Hungary, 4 University of Pécs, Medical School, Department of Biophysics, Pécs, Hungary, 5 Department of Optics and Quantum Electronics, University of Szeged, Szeged, Hungary, 6 University of Szeged, Department of Genetics, Szeged, Hungary

¤ Current address: Department of Bioengineering, Stanford University, Stanford, California, United States of America
* mihaly.jozsef@brc.hu

**Data Availability Statement:** All relevant data are within the manuscript and its Supporting information files.

## Abstract

During striated muscle development the first periodically repeated units appear in the pre-myofibrils, consisting of immature sarcomeres that must undergo a substantial growth both in length and width, to reach their final size. Here we report that, beyond its well established role in sarcomere elongation, the Sarcomere length short (SALS) protein is involved in Z-disc formation and peripheral growth of the sarcomeres. Our protein localization data and loss-of-function studies in the *Drosophila* indirect flight muscle strongly suggest that radial growth of the sarcomeres is initiated at the Z-disc. As to thin filament elongation, we used a powerful nanoscopy approach to reveal that SALS is subject to a major conformational change during sarcomere development, which might be critical to stop pointed end elongation in the adult muscles. In addition, we demonstrate that the roles of SALS in sarcomere elongation and radial growth are both dependent on formin type of actin assembly factors. Unexpectedly, when SALS is present in excess amounts, it promotes the formation of actin aggregates highly resembling the ones described in nemaline myopathy patients. Collectively, these findings helped to shed light on the complex mechanisms of SALS during the coordinated elongation and thickening of the sarcomeres, and resulted in the discovery of a potential nemaline myopathy model, suitable for the identification of genetic and small molecule inhibitors.

## Author summary

Sarcomeres are the smallest structural and functional units of muscles, characterized by a well-defined length and width in most muscle types. These two parameters are critically

**Funding:** This project was supported by the Hungarian Science Foundation (OTKA) (K132782 to J.M.) and (FK138894 to S.S.). The project was also supported by The National Laboratory of Biotechnology through the Hungarian National Research, Development and Innovation Office—NKFIH (grant No. 2022-2.1.1-NL-2022-00008 to J. M.), and by the 2022-2.1.1-NL-2022-00012 and TKP2021-NVA-19 projects with the support provided by the Ministry of Culture and Innovation of Hungary from the National Research, Development and Innovation Fund, financed under the 2022-2.1.1-NL and the TKP2021-NVA funding schemes (to E.M.).S.S. was supported by the János Bolyai Research Scholarship of the Hungarian Academy of Sciences and by the ÚNKP-22-5 New National Excellence Program of the Ministry for Culture and Innovation from the source of the National Research, Development and Innovation Fund, while M.A.T was supported by the University of Pécs, Medical School (KA-2023-12). The funders had no role in study design, data collection and analysis, decision to publish, or preparation of the manuscript.

**Competing interests:** The authors declare no competing financial interests.

important to determine the strength output of a muscle. Although previous studies described the contribution of several proteins involved in the regulation of sarcomere elongation and width, the mechanisms of these processes remained largely unclear. In this study we show that, beyond its well established role in sarcomere elongation, the SALS protein is involved in Z-disc formation and peripheral growth of the sarcomeres. Our protein localization data and loss-of-function studies in the *Drosophila* indirect flight muscle strongly suggest that radial sarcomere growth is initiated at the Z-disc. In addition, we demonstrate that the roles of SALS in sarcomere elongation and radial growth are both dependent on DAAM and Fhos, two formin type of actin assembly factors. Remarkably, we found that excess amounts of SALS promotes the formation of actin aggregates, similar to the ones described in nemaline myopathy patients. We expect that our studies will open new avenues of research to gain deeper insights into several key aspects of sarcomerogenesis, and might pave the way towards the development of novel myopathy models.

## Introduction

Myofibrils are composed of a regular array of sarcomeres, the basic structural and functional units of striated muscles. Each sarcomere is built up from three major filamental systems, the actin containing thin filaments, the Myosin containing thick filaments and the Titin containing elastic filaments connecting the thin and thick filament arrays. Together with several hundreds of associated proteins, the myofilaments arrange into a remarkably ordered pseudo-crystalline lattice in mature sarcomeres. Although muscle-specific variations do exist, the final size of the sarcomeres is tightly controlled as evidenced by the almost invariable length of the thin and thick filaments in relaxed skeletal muscles and the nearly constant sarcomere diameter in the muscle fiber. Owing to decades of research, we have a relatively good level of structural understanding of the sarcomeres, in particular with regard to the actin and myosin interactions [1–9]. However, much less is known about the molecular mechanisms behind the assembly and maturation of the sarcomeric units.

Structural organization of the sarcomeres is highly conserved across the animal kingdom, and due to these similarities, model organisms played an important role in elucidating the general principles of myofibril formation [10–13]. The *Drosophila* indirect flight muscle (IFM) became one of the most prominent model systems, where the power of genetics can be easily combined with biochemical approaches, molecular biology, microscopy and electrophysiology [14,15]. Development of the IFM begins in the early pupa when undifferentiated myoblasts fuse with each other to form myotubes. Subsequently, the myotubes attach to tendon cells at 12–16 hours after puparium formation (APF), which is followed by the formation of the first immature myofibrils that already contain periodic structures, the proto-sarcomeres, at 30 hours APF. During the remaining ~80 hours of pupal development the premyofibrils will fully mature that, of foremost, involves the assembly and growth of the sarcomeric units within them, until they reach their final length of 3.4 µm and diameter of 1.5 µm at 24 hours after eclosion [15–17].

Sarcomeric actin filaments, such as their non-muscle cell counterparts, exhibit a polarized morphology with a barbed or (+) end and a pointed or (-) end. Importantly, these filaments line up into uniformly registered arrays with their barbed ends at the Z-disc, and with their pointed ends flanking the central H-zone [18]. Former studies suggested that, unlike expected from analogies to non-muscle cells, length of the sarcomeric thin filaments is primarily regulated by a pointed end elongation mechanism [19,20]. The pointed end capping

Tropomodulin (Tmod) protein limits actin assembly at the pointed end [19,20], while the actin binding Wiskott-Aldrich syndrome homology 2 (WH2) domain-containing SALS protein was shown to promote thin filament lengthening by antagonizing the capping activity of Tmod by an unknown mechanism [21]. In addition to these two key players, the formin type of actin assembly factors DAAM, Fhos and Dia, Tropomyosin and the actin depolymerizing factor (ADF)/Cofilin are also implicated in the regulation of sarcomeric actin filament elongation (reviewed in [18]). Regarding the radial growth of the sarcomeres, Fhos was identified as a critical element [22], yet it remained unclear how Fhos mediates the incorporation of new peripheral actin filaments.

Here we present a structure-function analysis of the SALS protein comprising of two centrally located WH2 domains and a Proline-Rich (ProR) motif, that are flanked with extended N- and C-terminal intrinsically disordered regions without any obvious homology domains. Our results obtained in the IFM suggest that beyond its well established role in thin filament elongation [21], SALS is also involved in Z-disc formation, M-line organization and peripheral growth of the sarcomeres. We present a set of overexpression and genetic interaction studies demonstrating that SALS promotes both sarcomere elongation and radial growth in a formin dependent manner. In addition, we determined the developmental localization of the central region, and the N- and C-terminal tails of the SALS protein at the nanoscopic scale. These measurements revealed that at the end of pupal myofibrillogenesis SALS undergoes a conformational switch and it exhibits a reduced level at the pointed end, two changes that might be crucial to halt thin filament elongation in the young adults.

## Results

### SALS plays multiple roles in IFM development

The function of the *Drosophila* SALS protein has formerly been investigated mainly in larval somatic muscles and embryonic primary muscle cell cultures. These studies established a role in pointed end elongation [21], that has later been confirmed in the flight muscles as well [22]. Interestingly, beyond the evident role in sarcomere length regulation, the loss-of-function (LOF) analysis of *sals* mutant muscles revealed a more complex phenotype with occasional alterations in myofibril width and sarcomere organization [21]. To extend the functional characterization of SALS, we decided to take advantage of the extremely regular sarcomere organization of the IFM, ideally suitable for high resolution structural analysis. Because the null allele of *sals* is embryonic/larval lethal [21] the *UH3-Gal4* driver was used to knockdown the expression of *sals* specifically in the IFM, yielding a progeny with reduced flight ability both with a long (KK) and a short hairpin (TRIP) RNAi line (Fig 1A) (the former producing a stronger effect, and being selected for subsequent analysis). In line with former results, phalloidin staining of the mutant flight muscles 24 hours after eclosion (AE) revealed shorter sarcomeres and discontinuous thin filament edges at the pointed end (Fig 1B–1C", 1F and 1H, and S1 Fig). In addition, we noted a slight but highly reproducible change in sarcomere morphology. Normally, the IFM sarcomeres show a nearly perfect rectangular shape with a slight thickening at the M-line (Fig 1D–1D'), resulting in a 1.44 μm diameter at the Z-disc and 1.5 μm at the M-line. The uniform rectangular shape often turned into a hexagonal, sandglass-like form in the mutants (Fig 1E, 1E' and 1F), where we measured a diameter of 1.31 μm at the Z-disc and of 1.25 μm at the M-line, indicating a gradual sarcomere thinning from the Z-disc towards the M-line. Besides the moderate effect on sarcomere shape, by using the Z-disc marker α-Actinin we observed that ~37% of the Z-discs exhibited an impaired organization with altered shape and loss of perfect integrity (Fig 1C–1C" and 1G). Moreover, Obscurin staining revealed a distortion of the M-line structure in about 40% of the cases, often corresponding to sarcomeres

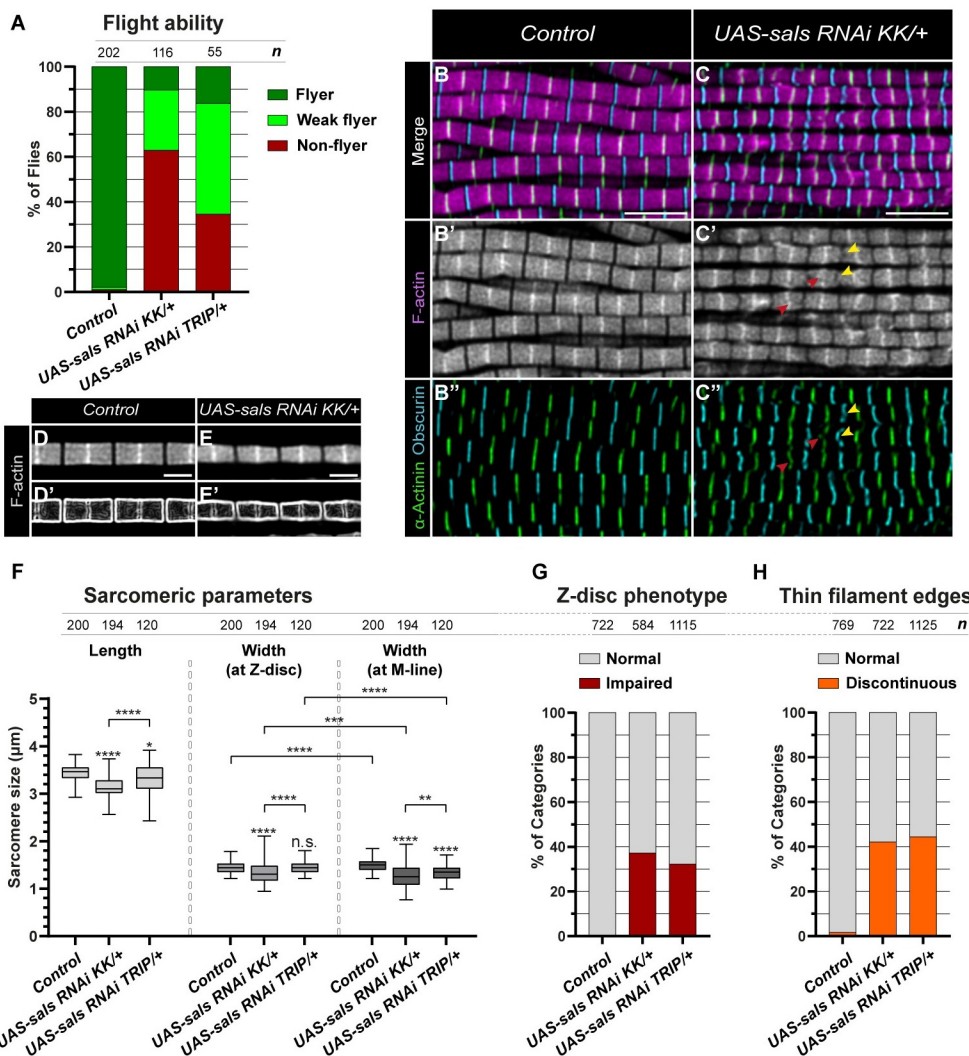

**Fig 1. The knockdown of *sals* impairs multiple aspects of sarcomerogenesis.** (A) Quantification of the flight ability of control (*UH3-Gal4/+ or -*) and *sals* knockdown flies (24h AE). *n* indicates the number of flies tested. (B-C") Comparison of the myofibrillar phenotype of muscles dissected from control (B-B") and *sals* knockdown (C-C") flies (24h AE). In *sals* mutant muscles a considerable number of Z-discs are fragmented or deformed (red arrowheads in C' and C") and many of the thin filament edges are irregularly shaped (yellow arrowheads in C' and C") as judged by F-actin (magenta in B, C; grey in B', C'), α-Actinin (green) and Obscurin (cyan) staining. (D-E') When stained for F-actin, sarcomeres of the control flies show a nearly perfect rectangular shape, which can be highlighted using ImageJ „Find edges" visual transformation (D'). In *sals* mutants the sarcomeres often display a sandglass-like morphology (E-E'). (F) Quantification of sarcomere length and width in control and in *sals* knockdown flies (24h AE). P-values were calculated using two-tailed unpaired Student's *t*-test with Welch's correction or Mann-Whitney *U* test according to the normality (***P≤0.001; ****P≤0.0001). (G-H) Quantification of the impaired Z-disc (G) and discontinuous thin filament edge (H) phenotypes. *n* indicates the number of sarcomeres (F), Z-discs (G) and M-lines (H) analyzed. Scale bars: 5 μm (B-C"), 2 μm (D-E').

where the thin filament pointed ends are not in perfect register with each other (Fig 1C–1C"and 1H).

To further confirm the phenotypic effects observed by confocal microscopy, we examined *sals* mutant muscles with transmission electron microscopy. In longitudinal sections of the adult IFM (24 hours AE) we found shorter sarcomeres, frayed myofilaments at the peripheral

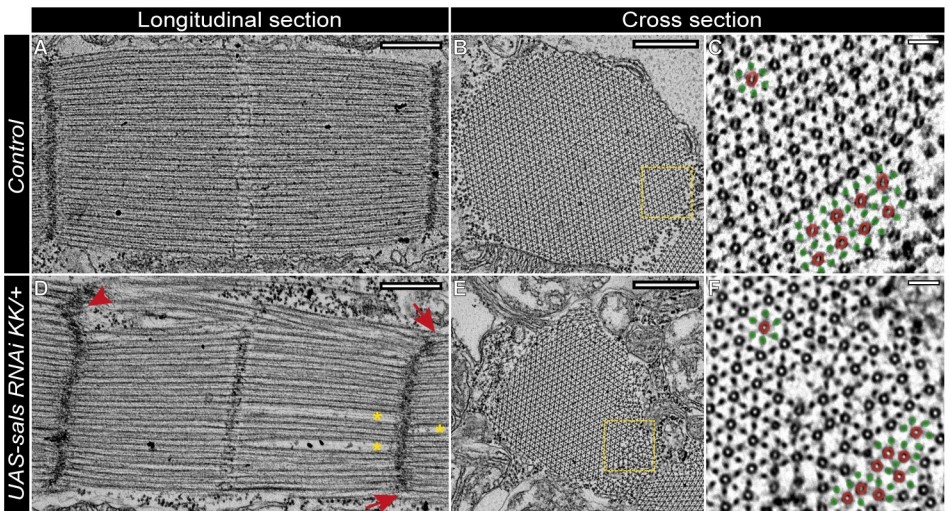

**Fig 2. TEM analysis of muscles from *sals* knockdown flies.** (A-F) TEM micrographs of sarcomeres dissected from control (*UH3-Gal4/-*) and *sals* knockdown flies (24h AE). (A) Longitudinal section of the control shows the highly ordered thin and thick filaments, and the straight, tightly packed Z-discs, while the *sals* mutant sarcomeres (D) often exhibit loosely organized filament sections (yellow asterisks), and deformed (red arrowhead) or shorter, centrally located Z-discs (red arrows). Cross section images of the control (B) and the *sals* mutant (E) IFMs reveal that the *sals* mutant sarcomeres are much thinner than the controls, and frayed at their periphery. In higher magnification, hexagonal lattice organization of the thick (dark orange ring) and thin (green circle) filaments is evident in the control sarcomeres (C, corresponding to the dashed square in B), whereas in *sals* mutants (F, dashed area in panel E) the hexagonal order and proper spacing of the filaments are often lost, in particular in the peripheral region of the myofibrils. Scale bars: 500 nm (A, B, D, E), 50 nm (C, F).

regions, and various Z-disc defects, such as broken or fragmented Z-discs and short, centrally located discs that failed to span the entire width of the sarcomeres (Fig 2A and 2D). The transverse sections revealed that many of the myofibrils are thinner than the wild type, and the hexagonal lattice organization is often impaired at their peripheral regions (Fig 2B, 2C, 2E and 2F). Together, these results clearly indicated that, although the most profound effect of SALS deficiency is the reduction in sarcomere length, it has multiple contributions to sarcomerogenesis in the IFM including Z-disc formation, M-line organization, and radial growth.

## Identifying the essential regions of SALS

To dissect the complex SALS requirements during IFM development, we initiated a structure-function analysis of the protein. To this end, we created a series of UAS-SALS transgenes to identify the functional contribution of the largely unstructured N-terminal region (aa. 1–345), the central Proline-Rich (aa. 345–379) and tandem WH2 regions (aa. 379–531), and the C-terminal domain (aa. 531–935) (Fig 3A). This set of transgenes was first probed in rescue experiments by using the muscle-specific *mef2-Gal4* driver and the transgenes in heterozygous condition in a *sals* null mutant background. Whereas the *sals* null mutants die during the embryonic (74%) or the L1 larval (26%) stages, expression of FL-SALS (Full Length SALS) was able to rescue this lethality till adulthood in ~74% of the progeny, while the remaining animals also reached to the L2-L3 or pupal stages (Fig 3B). Examination of the ΔProR-WH2, ΔWH2, ProR-WH2, N-term-ProR-WH2, N-term and C-term constructs revealed that these truncated forms are not able to provide a rescue till adulthood (Fig 3B), although the expression of these constructs resulted in larval lethality (L1/L2 stages), indicating a weak partial rescue. The ProR-WH2-C-term, WH2-C-term and ΔProR versions were clearly able to rescue the lethal

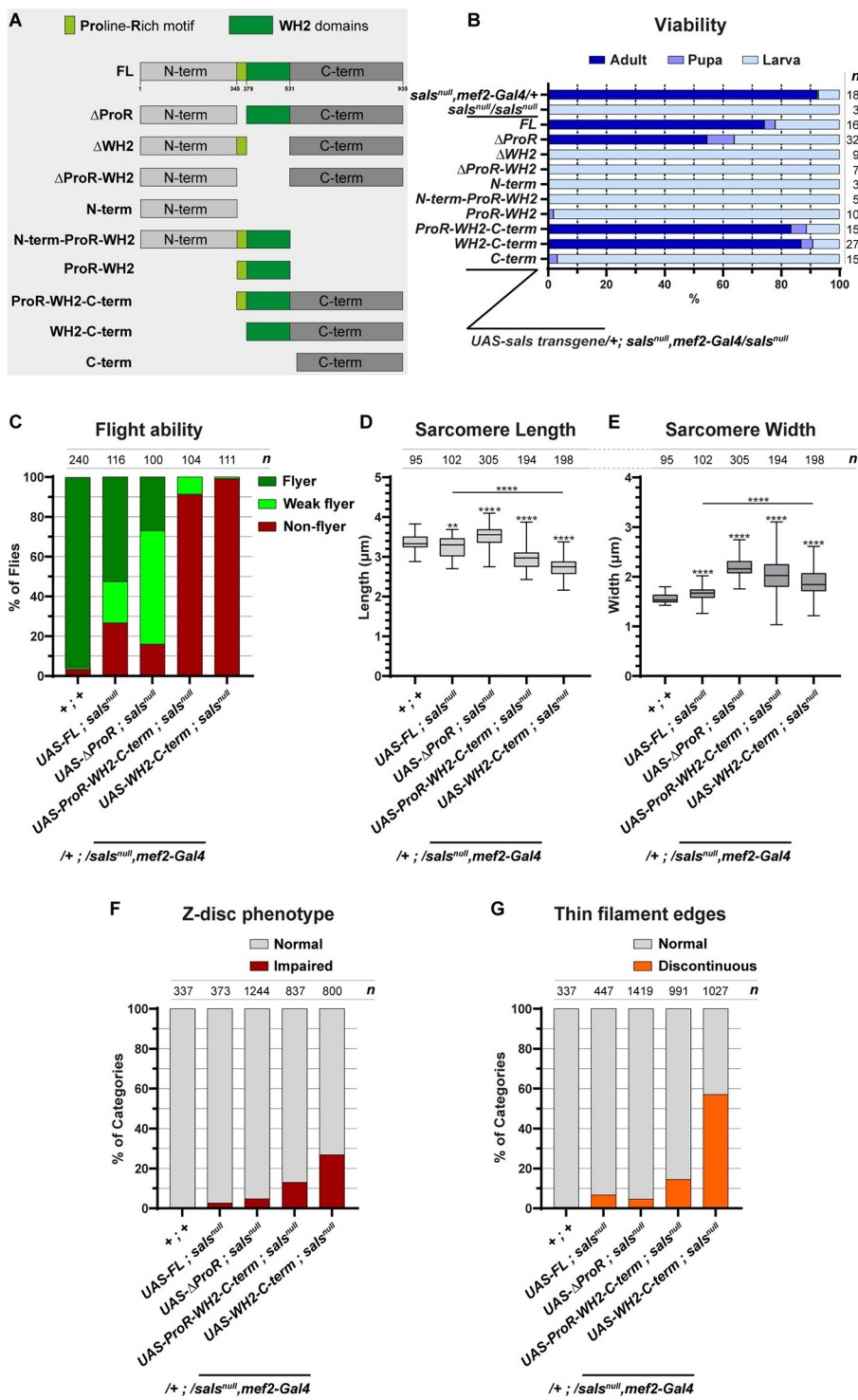

**Fig 3. Rescue ability of the truncated SALS proteins.** (A) Structure of the truncated SALS proteins, used for rescue and structure-function analysis. (B) The stacked row chart shows the rescue efficiency of the different SALS isoforms. Note that the *sals^null* homozygous mutants die as early larvae, and the rescue experiments were carried out in this genetic background with one copy of SALS transgene and one copy of the muscle-specific *mef2-Gal4* driver. *n* indicates the number of larvae examined. (C) Quantification of the flight ability of control (*sals^null*, *mef2-Gal4/+*) and rescued flies with the indicated genotypes (48h AE). *n* indicates the number of flies tested. (D-E) Quantification of sarcomere length (D) and sarcomere width (E) in control and rescued flies with the indicated genotypes (48h AE). P-values were calculated using two-tailed unpaired Student's *t*-test with Welch's correction or Mann-Whitney *U* test

according to the normality (\*\*P≤0.01; \*\*\*\*P≤0.0001). (F-G) Quantification of the impaired Z-disc (F) and discontinuous thin filament edge (G) phenotypes in control and in rescued flies. *n* indicates the number of sarcomeres (D-E), Z-discs (F) and M-lines (G) analyzed.

phenotype as ~53–86% of the larvae survived till adulthood (Fig 3B). These results suggest that the N-term and ProR regions are not essential for viability, while the central WH2 domains and the C-term region together are necessary for survival, as the sole presence of any of them is only sufficient for a weak partial rescue.

To address whether the ProR-WH2-C-term, WH2-C-term and ΔProR protein forms can completely rescue the lack of SALS, we compared the flight ability of the hatched adults to that of animals rescued with FL-SALS. As expected, the wild type protein was able to restore the flight ability in 73% of the progeny, however, flight ability of the ProR-WH2-C-term and WH2-C-term expressing adults was drastically reduced. In contrast, the ΔProR expressing adults showed only a moderate reduction of flight ability (Fig 3C). Because impaired flight is often associated with structural changes in the IFM, we next analysed IFM and sarcomere morphology in the rescued adults with confocal microscopy. These studies revealed that rescue with the FL form results in a slight reduction in sarcomeric length (3.23 μm), whereas in the cases of ProR-WH2-C-term and WH2-C-term, sarcomere length was significantly shorter than in the wild type (2.98 and 2.74 μm, respectively, versus 3.33 μm in the control) and the shortening was accompanied by an increase in diameter (2.04 and 1.89 μm, respectively, versus 1.53 μm in the control) (Fig 3D and 3E). In contrast, rescue by ΔProR resulted in an increase of both sarcomere length (3.50 μm) and width (2.20 μm) (Fig 3D and 3E). As to the Z-disc defects, the FL and ΔProR forms were able to largely rescue the impairments observed upon *sals* knockdown, while the ProR-WH2-C-term and WH2-C-term constructs provided a partial rescue in this regard (Fig 3F). With respect of the irregular thin filament edges at the M-line, the presence of the FL, ΔProR or ProR-WH2-C-term forms was sufficient for wild type development in most animals, but in the case of the WH2-C-term protein 57% of the M-lines remained impaired (Fig 3G). Because the transgenic lines generated for the structure-function analysis all carry a 3xFlag-tag at an N-terminal position, we have been able to determine the sarcomeric protein distribution pattern of 4 SALS protein forms providing a rescue till adulthood. Of these, the Flag-tagged FL-SALS displayed a strong accumulation in the H-zone and a weaker staining at the Z-disc of the IFM myofibrils, which highly resembles the expression pattern of the wild type endogenous protein (S2 Fig). The ΔProR version exhibited a similar pattern to that of FL-SALS, whereas the ProR-WH2-C-term and WH2-C-term forms strongly accumulated in the H-zone but their enrichment at the Z-disc was much weaker than in the case of FL-SALS (S2 Fig). Thus, the truncated proteins providing the best rescue (i.e. till adulthood) kept the ability to integrate into the myofibrils in largely comparable amounts as the wild type protein. Despite this, localisation of the ΔProR form is closer to wild type than that of the ProR-WH2-C-term and WH2-C-term forms, which might explain that ΔProR provides a significantly better rescue of the flight than the other two truncated forms (Fig 3C).

Based on the data collected from the rescue experiments (summarized in S4 Fig) we conclude that, unlike the ProR motif, the central WH2 region of the protein, formerly implicated in G- and F-actin binding [23], is clearly essential for SALS function. In addition, the C-terminal arm is also indispensable for viability. In the absence of these domains, the *sals* mutants die during the larval stages, presumably due to muscle defects indicated by the slow crawling movement of the mutant larvae. Interestingly, the WH2 and the C-terminal regions together are not only able to rescue the lethality of *sals*, but muscle-specific expression of the ProR-WH2-C-term or WH2-C-term proteins is sufficient to ensure a largely normal

sarcomere development in the IFM. Nevertheless, the flight ability of these rescued animals is severely reduced, and their sarcomeres are shorter than in the wild type often displaying minor irregularities in the thin filament order both at the Z-disc and in the H-zone, suggesting that the N-terminal region is important for thin filament extension and/or length regulation, as well as for proper Z-disc formation.

## The overexpression of SALS confirms the role in elongation and peripheral growth of the sarcomeres, and reveals a myopathy-related phenotype

To explore further the functional abilities of the truncated SALS versions, we tested their phenotypic effects upon overexpression in a wild type background. When expressed with the muscle-specific *mef2-Gal4* driver, the presence of the truncated SALS proteins had no effect on viability, however, flight was clearly impaired by them, even in the case of the FL-SALS wild type control (Fig 4A). Excess of ΔWH2, ΔProR-WH2, WH2-C-term or C-term has completely abolished flight ability, whereas the rest of the tested transgenes reduced flight in about 50% of the flies (Fig 4A). Subsequently, we focused on the non-flyer category, and examined the effects at the myofibrillar and sarcomeric level. Although none of the constructs affected gross myofibrillar organization in the IFM (with the exception of ΔWH2 that induced irregularities in myofibril arrangement; Fig 4B–4D), all of them induced changes in sarcomere length and/ or width. The most obvious effect on sarcomere length regulation was that ΔWH2, N-term, N-term-ProR-WH2, ProR-WH2, ProR-WH2-C-term or WH2-C-term overexpression resulted in the formation of shortened sarcomeres (Fig 4E). In contrast, FL-SALS and C-term had a very mild or no significant effect on sarcomere length, only ΔProR and ΔProR-WH2 were able to slightly increase sarcomere length (3.52 μm and 3.62 μm, respectively). These data are consistent with the notion that SALS affects thin filament elongation, and it appears that most of the truncated versions exhibit a dominant negative effect on sarcomere length, possibly indicating that SALS either acts as a multimer or the binding partners of SALS are present in stoichiometrically strictly controlled amounts. With regard to sarcomere width, we found that expression of FL-SALS, ΔProR, ΔProR-WH2, N-term-ProR-WH2 and ProR-WH2 strongly increased the diameter of the sarcomeres (Fig 4F). The N-term, ProR-WH2-C-term and C-term proteins induced a moderate increase, the WH2-C-term had no significant effect, while ΔWH2 decreased the width of the sarcomeres (Fig 4F). Thus, with the exceptions of WH2-C-term and ΔWH2, the truncated SALS constructs promoted the radial growth of the sarcomeres, and this effect was often more pronounced than the effects on sarcomere length regulation. Given that the loss of SALS function results in thinner myofibrils [21], we interpret the increase in sarcomere width as a gain of function effect revealing an important role in radial growth. The strongest effect on sarcomere width was caused by the ΔProR version which showed a 50% increase (Fig 4F). As compared to this, FL-SALS had a weaker (34.2% increase), yet still a robust effect, suggesting that, in context of the entire protein, the ProR motif might play a regulatory role of radial growth. If so, this is likely to be mediated by interacting with an SH3 domain protein [24–27], as we showed that the SALS ProR region does not interact with Profilin [23]. The second strongest effect on sarcomere width was exhibited by the isolated central region of the protein, expressed by the ProR-WH2 construct. Whereas the N-term and C-term regions alone induced only a modest increase in width, when the ProR-WH2 region was extended by either of these regions, the robust effect of the isolated central domains was moderately (N-term-ProR-WH2) or strongly (ProR-WH2-C-term) suppressed (Fig 4F). Collectively, these results indicate that the WH2 domains of SALS are critically required to promote radial sarcomere growth, however, this activity appears to be regulated in a complex manner. This regulation involves the N- and/or C-term regions that may act due to

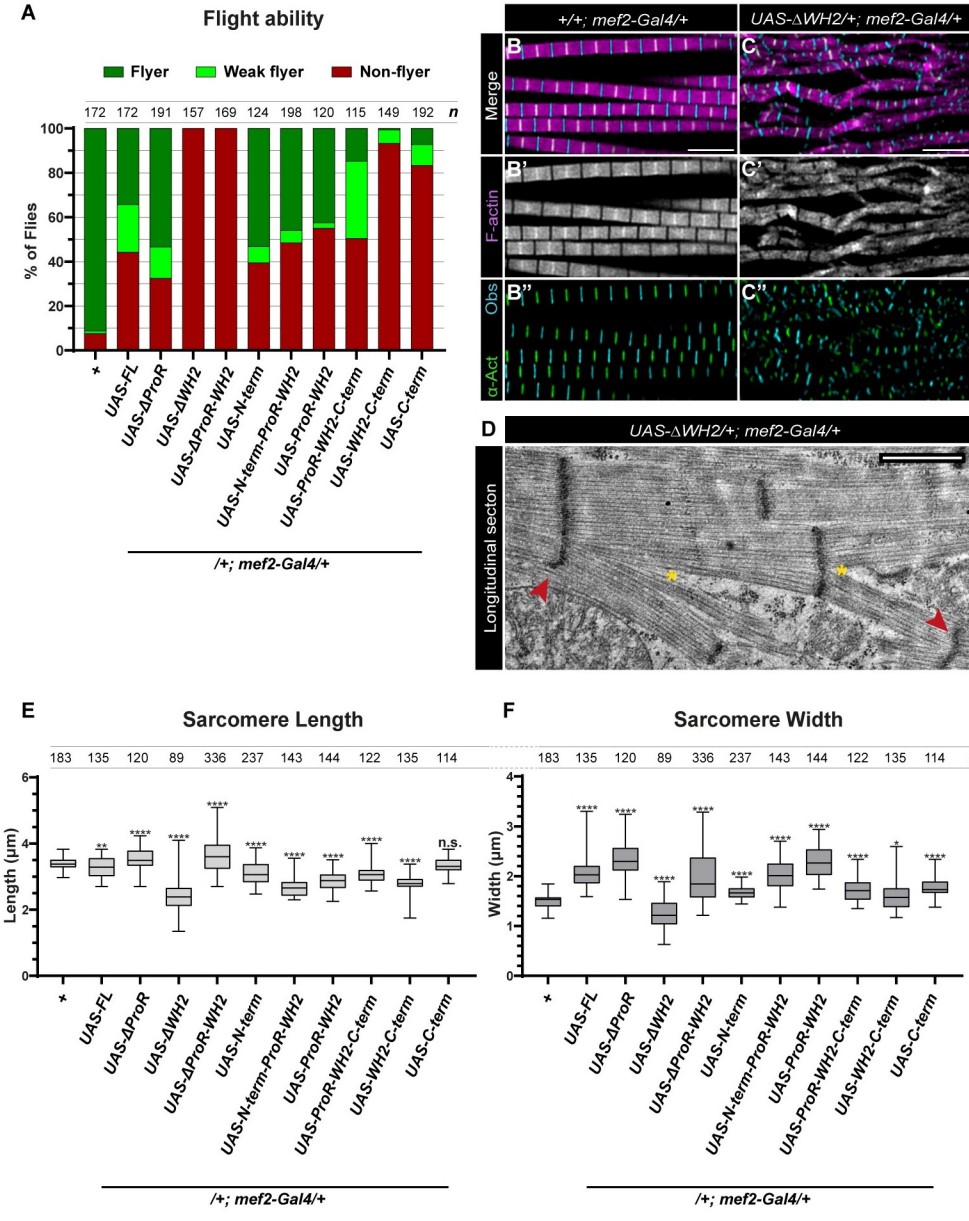

**Fig 4. Overexpression of SALS reveals a role in the peripheral growth of the sarcomeres.** (A) Quantification of the flight ability of control (*mef2-Gal4/+*) flies and that of expressing the indicated SALS protein form (in one copy with *mef2-Gal4*) (24h AE). *n* indicates the number of flies tested. (B-C") Comparison of the myofibril morphology of control (B-B") muscles with muscle expressing the ΔWH2 truncated SALS protein (C-C"). Excess of the truncated protein severely disrupts sarcomere organization indicated by F-actin (C') (magenta in B, C; grey in B', C'), α-Actinin (C") (green) and Obscurin (cyan) staining. (D) TEM analysis of the myofibrillar organization upon ΔWH2 overexpression. The presence of shorter, centrally located or impaired Z-discs (red arrowhead) and split myofibrils (yellow asterisks) is evident. Quantification of the sarcomere length (E) and sarcomere width (F) measurements in flies expressing the indicated SALS protein (24h AE). P-values were calculated using two-tailed unpaired Student's *t*-test with Welch's correction or Mann-Whitney *U* test according to the normality (n.s., not significant P>0.05; *P≤0.05; **P≤0.01; ****P≤0.0001). *n* indicates the number of sarcomeres analyzed (E-F). Scale bars: 5 μm.

intramolecular mechanisms or by the recruitment of their interaction partners capable of modulating WH2/SALS activity.

Besides the alterations in the basic shape of the sarcomeres, all constructs induced the formation of actin-rich aggregations, mostly in the extra-myofibrillar space in the peripheral region of the myofibers (S3G–S3J Fig). The aggregates were sometimes attached to the myofibrils, and depending on the construct, they varied in size and organization. The FL-SALS and ΔProR constructs promoted the formation of large, loosely organized, actin-based meshworks covering huge areas over the myofibrils. On the other extreme, the N-term and C-term constructs induced the occurrence of a few, compact actin clamps, while the rest of the constructs exhibited less compact aggregates with medium frequency (S3K Fig). In addition, in the presence of ΔProR, ΔWH2, ΔProR-WH2, ProR-WH2 or N-term, we detected huge, actin-rich, ring-like structures (Fig 5A–5C"), highly resembling the giant Z-disc like structures described in some of the nemaline myopathy (NM) cases [28,29]. Just as in NM, most of these assemblies were connected to a native Z-disc (Fig 5A–5A") or interconnected several myofibrils (Fig 5B–5B"), although we frequently found ones with less obvious myofibril association as well (Fig 5C–5C"). The ΔWH2 construct induced the highest occurrence of the giant Z-discs that could be detected in nearly all individuals, typically with 13–22 such assemblies in the adult muscles (Fig 5E–5G). Beyond the giant Z-discs, this construct also induced the formation of different other types of actin aggregations of various size and shape (including zebra bodies often present in myopathy patients) (Fig 5D and S3 Fig). By using a set of Z-disc markers, we revealed that the central actin bundle of the giant Z-discs is flanked by Kettin, α-Actinin and Zasp52, further confirming the similarity with Z-discs (Fig 5K–5M"). Immunostaining of the pupal IFM revealed that the actin aggregations, including the giant Z-discs, appear already at 30–48 APF and they remain detectable till adulthood (Fig 5N–5Q) with a similar protein composition as their adult counterparts (Fig 5H–5M"), but without a significant increase in their number over time (Fig 5G). Importantly, we could detect the presence of the Flag-tagged, overexpressed proteins both in the giant Z-discs and the actin aggregates (S3A–S3F" Fig), which suggests that SALS has the ability to directly impact actin filament formation, even in an ectopic, extra-myofibrillar situation.

In summary, these overexpression studies further corroborated the role of SALS in sarcomere/thin filament elongation, and highlighted the importance of this protein in radial sarcomere growth. Additionally, nearly all constructs were able to induce the formation of extra-myofibrillar actin aggregations upon overexpression (S3G–S3J and S4 Figs), and we noted the formation of giant Z-disc like structures and zebra bodies in several cases. Despite their considerable morphological variability, these actin-based protein accumulations are remarkably similar to the various actin rods and thin filament arrays found in NM patients [28,29]. Collectively, these results support that SALS is a central regulator of thin filament assembly in *Drosophila* muscles. Whereas the role in pointed end elongation was already well established, the findings on radial growth and Z-disc formation shed new light on two other important aspects of sarcomere development. Although we cannot exclude that these processes require separate functions of SALS, taking the overexpression data together with our LOF analysis, we think that these findings are most consistent with the view that radial sarcomere growth is intimately linked to Z-disc growth. Therefore, it is tempting to speculate that the roles of SALS in radial sarcomere growth and Z-disc organization are connected to each other.

## SALS regulates sarcomere size in a formin dependent manner

Members of the formin protein family promote the formation of unbranched actin filaments, and these proteins were shown to be required for sarcomere formation in various model

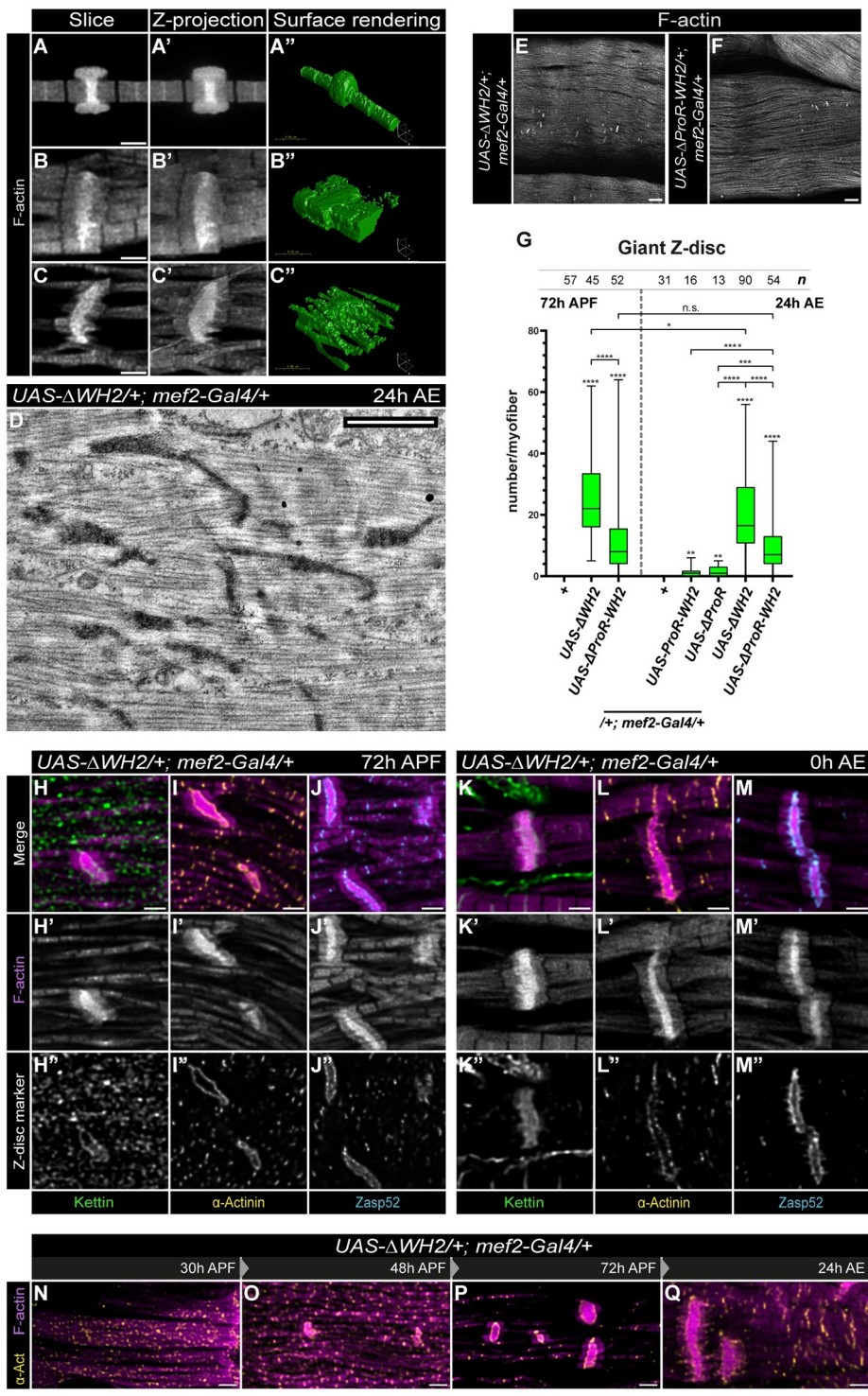

**Fig 5. The excess level of SALS induces the formation of giant, Z-disc-like, actin structures.** (A-C") 2D representation (A-A', B-B', C-C') and 3D visualization (A", B", C") of three different examples of the giant Z-disc-like actin structures observed upon SALS overexpression. (D) A TEM image demonstrating the presence of giant Z-disc-like structures and zebra bodies upon ΔWH2 overexpression. (E-G) Frequency of the giant Z-discs varies depending on the truncated SALS version used, the ΔWH2 form being the most potent in this context, as quantified in G. *Mef2-Gal4* / + flies were used as control. P-values were calculated using Mann-Whitney *U* test. (**P≤0.01; ***P≤0.001; ****P≤0.0001). *n* indicates the number of muscle fibers analyzed. (H-M") Confocal microscopy images of large, rod-like actin structures in pupal (H-J") and young adult (K-M") muscles expressing the ΔWH2 protein. Beyond

F-actin (H', I', J', K', L', M'), these giant actin aggregations contain several Z-disc proteins as well, such as Kettin (H", K"), α-Actinin (I", L") and Zasp52 (J", M"). F-actin staining is in magenta, Kettin is in light green, α-Actinin is in yellow and Zasp52 is in cyan in the merged images. (N-Q) Developmental analysis of the giant Z-discs, revealing that they are present already at 48h APF, and that their size appears to grow during the subsequent stages. Scale bars: 2 μm (A-A', B-B', C-C', H-M", N-Q), 4 μm (A", B", C"), 1 μm (D), 20 μm (E-F).

systems [22,30–35]. In the case of the IFM, 5 of the 6 *Drosophila* formins are expressed during IFM development [15], and a functional link has been made for DAAM, Fhos and Dia [22,31,33]. Of these, we focused on DAAM and Fhos, exhibiting the strongest defects in sarcomerogenesis, including the formation of shorter and thinner sarcomeres than in wild type [22,33], suggesting that DAAM and Fhos are required both for sarcomere elongation and peripheral growth. Given that the SALS protein has similar properties, next we used a genetic interaction approach to address whether SALS regulates sarcomere size in a formin dependent manner.

During these experiments we worked with two SALS protein forms, FL-SALS and ΔProR. Of these, the overexpression of FL-SALS has a negligible effect on sarcomere length, whereas the ΔProR construct significantly increases it (Fig 4E). At the same time, both protein forms increase sarcomere width, although ΔProR has a stronger effect (Fig 4F). We asked whether reducing the level of DAAM or Fhos by RNAi mediated silencing affects the phenotypes caused by SALS overexpression. Muscle-specific knockdown of these formins with *mef2-Gal4* reduces both sarcomere length and width (Fig 6A–6L), and we found that it suppressed the effect of FL-SALS and ΔProR overexpression (Fig 6M–6P). Sarcomere length was reduced to the degree when these formins are knocked down in a wild type background (Fig 6M and 6N), suggesting that the increase in length induced by SALS is entirely formin dependent. The silencing of DAAM or Fhos has also reduced the width of the sarcomeres observed upon FL-SALS and ΔProR overexpression (Fig 6O and 6P). However, while the knockdown of Fhos resulted in a complete suppression to the level when Fhos is silenced in a wild type background (Fig 6P), the DAAM knockdown had a partial effect (Fig 6O). To further corroborate these findings, we also examined the effect of a *DAAM* and *fhos* null allele, $DAAM^{Ex68}$ and $fhos^{\Delta 1}$ [22,36]. When used in heterozygous conditions, $DAAM^{Ex68}$ suppressed the increase of sarcomere length induced by FL-SALS, while the *fhos* allele had no effect under these conditions (S5A and S5B Fig). At the same time, both alleles were able to partly suppress the increase of sarcomere width caused by FL-SALS (S5C and S5D Fig). In addition, we also revealed that the flightless phenotype induced by FL-SALS overexpression is also significantly suppressed by $DAAM^{Ex68}$ and $fhos^{\Delta 1}$ (S5E Fig). Thus, these data suggest that the SALS protein and the two formins (DAAM and Fhos) work together to promote thin filament elongation as well as the peripheral growth of the sarcomeres.

Unlike the formin proteins, which act as positive regulators of thin filament elongation, Tmod is thought to block actin filament elongation by antagonizing the effect of SALS in primary muscle cells [21]. To probe this converse effect in the IFM, we examined the phenotypic consequences of Tmod overexpression in the flight muscles (with the help of the *UH3-Gal4* driver). The excess amount of Tmod caused a plethora of myofibrillar defects, including Z-disc and M-line destructions (as judged by Kettin and Obscurin staining), loss of uniform thin filament length, accompanied by a slight reduction in the length (3.19 μm versus 3.37 μm in controls) and width of the sarcomeres (1.37 μm versus 1.42 μm in controls) (Fig 7A–7F). When FL-SALS is expressed with the *UH3-Gal4* driver, sarcomere length is similar to the control, while width, measured at the Z-disc, is clearly greater than in the control samples (1.62 μm versus 1.44 μm) (Fig 7E and 7F). Upon the concomitant expression of Tmod and

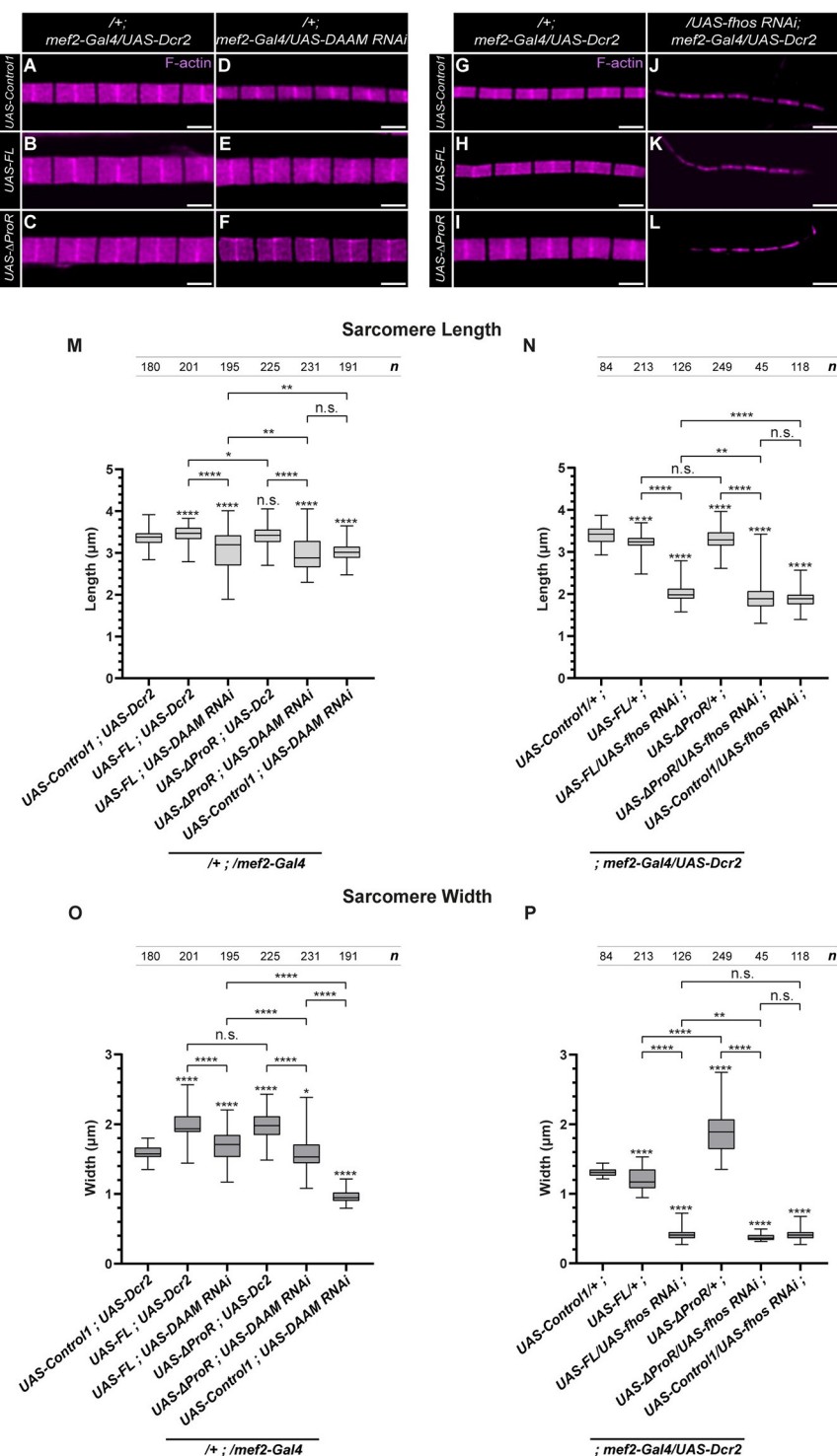

**Fig 6. SALS regulates sarcomere growth via formin interactions.** (A-F) F-actin staining of myofibrils dissected from FL-SALS and ΔProR expressing IFMs in a *mef2-Gal4/UAS-Dcr2* background (B, C) and upon *DAAM* knockdown (E, F) (24h AE). The controls used are *UAS-Control1/+; mef2-Gal4/UAS-Dcr2* in A, and *UAS-Control1/+; mef2-Gal4/ UAS-DAAM^RNAi* in D. (G-L) F-actin staining of myofibrils dissected from FL-SALS and ΔProR expressing IFMs in a *mef2-Gal4/UAS-Dcr2* background (H, I) and upon *fhos* knockdown (K, L) (96h APF). The controls used are *UAS-Control1/+; mef2-Gal4/UAS-Dcr2* in G, and *UAS-Control1/UAS-fhos^RNAi; mef2-Gal4/UAS-Dcr2* in J. As *UAS-Control1* we used *UAS-Vang*, a transgenic line with no significant effect on IFM development (for further details see M&M). (M-P) Quantification of sarcomere length (M, N) and sarcomere width (O, P) in mutant flies with the

indicated genotype. The M and O panels refer to adult IFMs (24h AE), whereas the N and P panels apply to pupal IFMs (96h APF). P-values were calculated using two-tailed unpaired Student's *t*-test with Welch's correction or Mann-Whitney *U* test according to the normality (n.s., not significant P>0.05; **P≤0.01; ****P≤0.0001). *n* indicates the number of sarcomeres measured. Scale bars: 2 μm.

FL-SALS (Fig 7G–7I) we measured the sarcomere length as 3.28 μm, whereas the width was 1.88 μm (Fig 7E and 7F). In this manner, these observations further confirmed the antagonistic activities of Tmod and SALS on thin filament elongation. Moreover, consistent with findings in vertebrate [37] and in *Drosophila* larval body wall muscle models [38], they also indicate

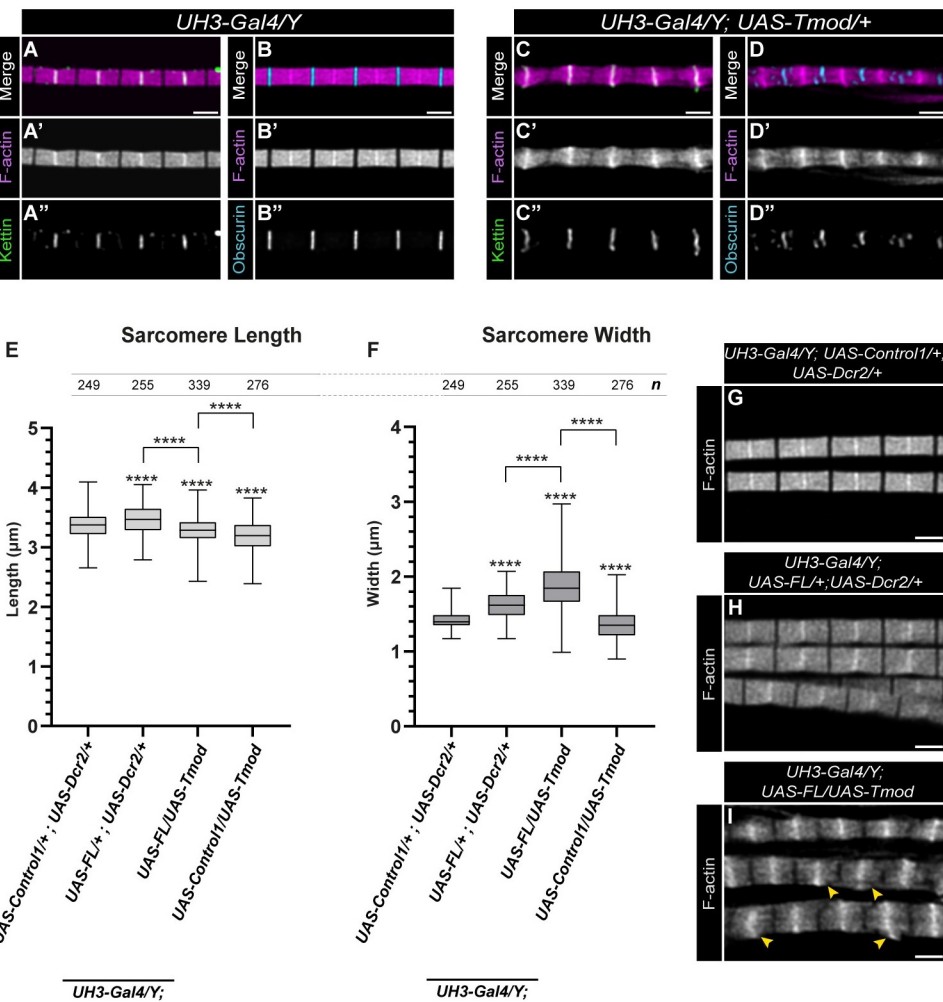

**Fig 7. SALS and Tmod interaction during peripheral sarcomere growth.** (A-B") Control (*UH3-Gal4/Y*) myofibrils display a stereotypic, rectangular sarcomere morphology, with well-organized Z-discs and M-lines, as judged by Kettin (A-A") and Obscurin (B-B") staining, respectively. (C-D") Upon overexpression of Tmod, the sarcomeres exhibit a sandglass-like morphology, and partly disrupted Z-discs (marked with Kettin) (C-C") and M-lines (marked with Obscurin) (D-D"). (E-F) Quantification of the length (E) and width (F) of the sarcomeres in muscles expressing SALS or Tmod, or both together. Note that *UAS-Vang* was used as *UAS-Control1*. P-values were calculated using two-tailed unpaired Student's *t*-test with Welch's correction or Mann-Whitney *U* test according to the normality (n.s., not significant P>0.05; *P≤0.05; ****P≤0.0001). *n* indicates the number of sarcomeres measured. (G-I) As compared to the control myofibrils (G), the IFM-specific expression of FL-SALS does not lead to major morphological alterations (H), while the concomitant expression of SALS and Tmod (I) results in the formation of thicker myofibrils with shortened sarcomeres often exhibiting wider Z-discs and thin filaments that are not in register with each other (yellow arrowheads). Scale bars: 2 μm.

that Tmod is likely to play multiple roles during IFM development, including a direct or indirect contribution to the regulation of radial sarcomere growth.

## Nanoscopic analysis of SALS distribution during IFM development reveals a major conformational change of SALS after the completion of sarcomere elongation

Previously, we performed a nanoscopic analysis in the IFM of adult flies (24h AE) [39], and by using an antibody specific for the central region of SALS [21], we found that SALS displayed a "double-line type" distribution by forming parallel lines both in the Z-discs and H-zone of the sarcomeres. We measured an average Z-line peak distance of 46.7 nm, positioning it close to the thin filament barbed ends, while in the H-zone we found a median M-line peak distance of 64 nm, placing it into the immediate vicinity of the pointed end. To extend these studies, and address the developmental roles of SALS, we decided to investigate the nanoscale distribution of the protein during pupal development. We tested the nanoscopic localizations in individual myofibrils isolated from four different timepoints, covering the transition between the growing and mature phases of sarcomerogenesis. Whereas the accumulation of SALS in the Z-discs of growing sarcomeres was too weak to enable a STORM analysis with sufficient precision, the H-zone signal was suitable for high resolution studies. By using the pointed ends as reference points (marked by Tmod) in the H-zone, we found that the width of the H-zone is significantly wider in growing sarcomeres (165 nm) (72 hours APF) than in mature sarcomeres (123.4 nm) (24 hours AE; Fig 8A and 8B). While in the mature sarcomeres Tmod and SALS practically 'colocalize', in the earlier, growing phase we found a substantial difference, as the central region of SALS localized significantly closer to the M-line, suggesting thin filament-independent localization. Curiously, in stretched myofibrils both the SALS and Tmod patterns are stretched, demonstrating that both proteins are connected either directly or indirectly to the thin filament array (Fig 8C). To clarify this apparent contradiction, and to get a more comprehensive picture of the nanoscopic organization of SALS, we also determined the localization of its N- and C-termini. For these measurements we expressed FL-SALS protein variants Flag-tagged either on the N- or C-terminus, and we mapped the exact position of the Flag-tag in growing (72 hours APF) and mature (24 hours AE) sarcomeres. We first determined the localization of the N-terminal Flag-tag and found that, as compared to the central WH2 domain containing region, it is located closer to the pointed ends in growing sarcomeres, while in mature sarcomeres it is closer to the center of the H-zone (Fig 9A and 9B). Contrasting that, the C-terminal Flag-tag maintains its localization in close proximity to the pointed ends both in growing and mature sarcomeres (Fig 9A and 9B), which suggests a continuous interaction between the pointed ends and that of C-term SALS. Given that the central WH2 domain-containing region is associated with the pointed ends only in mature sarcomeres (Fig 9B), it might remain free in the growing sarcomeres, possibly to fulfill essential functions in thin filament elongation.

These measurements also demonstrated, that in line with its proposed role as a promoter of thin filament elongation, the conformation of SALS changes significantly as thin filament elongation halts at the end of myofibrillogenesis. This raises the possibility that the proper length of thin filaments may be directly regulated by the conformational changes of the SALS protein. However, based on the dSTORM images and the measured localization densities (see Materials and methods), the relative amount of SALS appears to decrease significantly, while the relative amount of Tmod remains constant during the progression of myofibrillogenesis (Fig 9C). This quantitative change can be an additional level of regulation that could explain the shift in thin filament dynamics at the end of myofibrillogenesis.

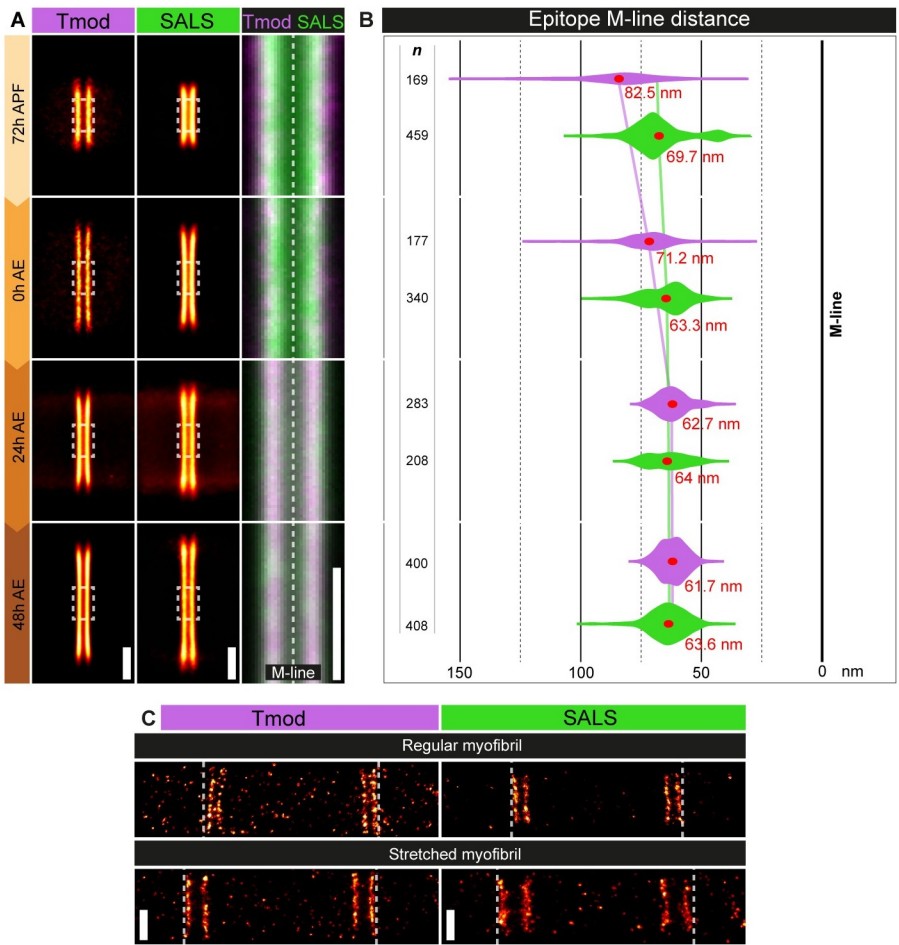

**Fig 8. Nanoscopic analysis of SALS and Tmod distribution during IFM development.** (A) Distributional pattern of Tmod (first column) and SALS (second column) in the H-zone visualized at four different timepoints of muscle development (72h APF, 0h AE, 24h AE, 48h AE). The third column (magnification and combination of the dashed squares in the adjacent images) shows the nanoscale distribution of Tmod and SALS near the M-line. Tmod staining is in magenta, SALS is in green (stained with an antibody specific for the central region of the protein). (B) Violin plot representation of the longitudinal epitope distributions of Tmod (magenta) and SALS (green) relative to the M-line. Red dots and numbers show the mean values. *n* indicates the number of H-zones analyzed. (C) Comparison of the localization pattern of Tmod (left panels) and SALS (right panels) in regular and stretched myofibrils. Dashed lines highlight the degree of shift upon stretching. Scale bars: 500 nm.

Nanoscale determination of the positions of the central WH2, and the N- and C-terminal regions of SALS during pupal development has clearly revealed that the relative order of these domains changes as compared to each other and to that of the thin filament pointed ends (Fig 9B). Thus, SALS is very likely to undergo a major conformational change after the completion of thin filament elongation, and in parallel to that, it largely diminishes from the H-zone. This indicates that SALS is required for pointed end elongation, but it is unlikely to play a role in the maintenance of thin filament length, which is entirely consistent with our finding that the adult stage specific knockdown of *sals* has no effect on IFM function (S6 Fig).

## Discussion

In this study, we report the structure-function analysis of SALS, one of the few proteins implicated in pointed end elongation of the sarcomeric thin filaments. Using the *Drosophila* IFM as

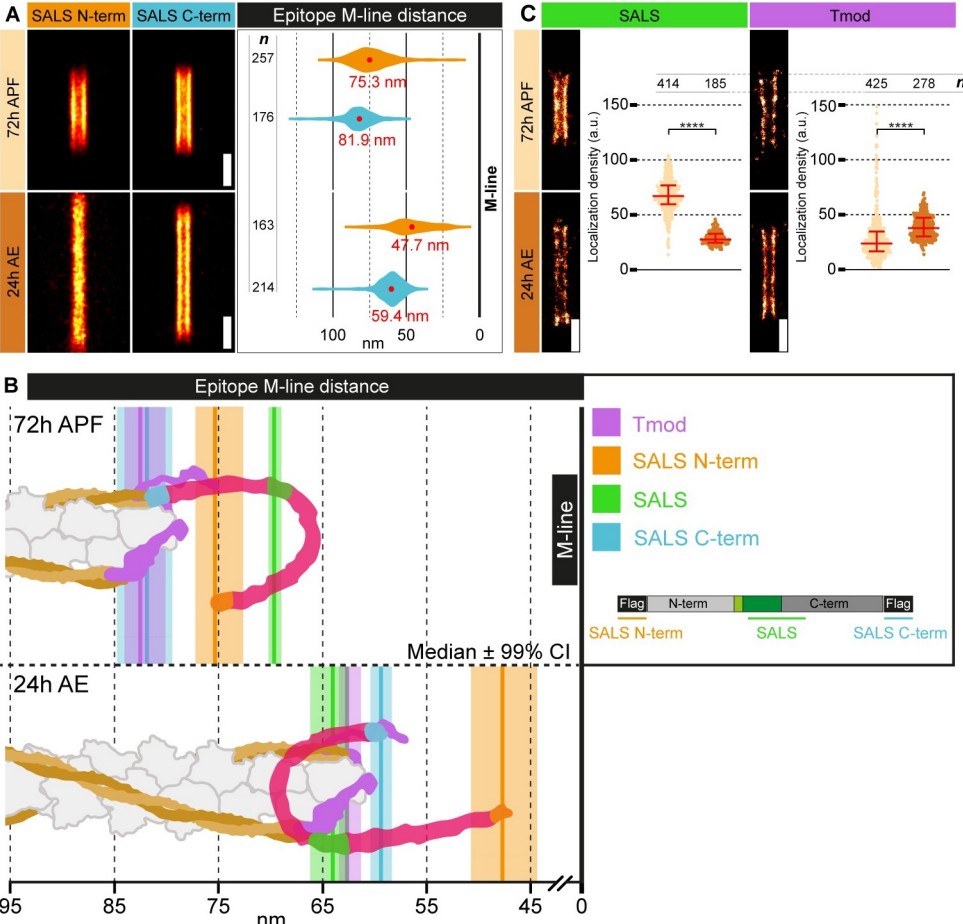

**Fig 9. Nanoscopic reconstruction reveals the conformational change of SALS at the end of myofibrillogenesis.** (A) Distributional pattern of an N-terminally and a C-terminally located epitope of SALS in the H-zone of a growing (72h APF) and a mature (24h AE) sarcomere. Violin plot shows the longitudinal epitope distributions related to the M-line (N-term in dark orange, C-term in cyan). Red dots and numbers show the mean values. (B) Schematic drawing of a thin filament pointed end of a growing (72h APF) and a mature (24h AE) sarcomere indicating the position of Tmod and three SALS epitopes. Tmod is in magenta, SALS N-term is in dark orange, SALS central region is in green and SALS C-term is in cyan. Note the change of the relative order of these epitopes as compared to each other, and that of the M-line. (C) Visualization and quantification of the localization density of SALS and Tmod in the H-zone of a growing (72h APF) and a mature (24h AE) sarcomere. P-values were calculated using Mann-Whitney *U* test. (****P≤0.0001). *n* indicates the number of H-zones analyzed. Scale bars: 500 nm.

our major model system, we extended the LOF analysis of *sals*, leading to the conclusion that the role of SALS is not restricted to pointed end regulation (i.e. an H-zone function), instead the protein is crucial for Z-disc formation and radial growth of the sarcomeres. Rescue experiments revealed that the WH2 domains and the C-terminal region are required both for sarcomere elongation and radial growth, whereas the unstructured N-terminal region is important for length determination and the ProR motif is involved in proper regulation of the radial growth. Examination of the genetic hierarchy of *sals* demonstrated that SALS works together with two formins, DAAM and Fhos, to support actin elongation and peripheral sarcomere growth. In addition to these studies, we carried out a developmental analysis of SALS localization in the IFM at the nanoscale level, and discovered that at the end of sarcomerogenesis

SALS is subject to a conformational change that might allow higher affinity Tmod binding at the pointed end, thereby contributing to the termination of thin filament elongation.

Previous studies reported that in the developing IFM the SALS protein is enriched in the H-zone, and also at the Z-disc [21,39]. The H-zone localization is entirely consistent with its well established role in pointed end elongation, however, functional importance of the Z-disc accumulation remained largely elusive. By discovering a role in Z-disc formation, our new findings provide an explanation for both major elements of the localization pattern. In addition, as a third function, we found a requirement in radial growth of the sarcomeres. In the case of Z-disc formation, we show that upon *sals* knockdown the myofibrils often exhibit impaired, fragmented or broken Z-discs, whereas overexpression of some of the truncated versions induces the formation of huge Z-disc-like structures, as judged by their organization and the presence of actin and several other known Z-disc proteins. Regarding the radial growth function, overexpression of either the full length or certain truncated versions of SALS drastically increases sarcomere width, suggesting that the excess and/or the ectopic activation of the protein is sufficient to promote peripheral filament addition. As compared to this, when *sals* function is compromised during IFM development, a subpopulation of the myofibrils exhibits sandglass-shaped sarcomeres with gradually decreasing sarcomere width from Z-disc to M-line. Because the RNAi knockdown of *sals* only partly reduces the protein level [21], we cannot exclude that a complete depletion of the protein would result in the formation of uniformly thinner sarcomeres in the IFM (similarly to the *sals* mutant embryonic muscles). Surprisingly, a former study, in which Shwartz et al. investigated GFP::Actin88F incorporation in the young adult stage after *sals* knockdown during the second half of pupal development [22], reported a normal peripheral thickening of the sarcomeres in these conditions. Since the knockdown of *sals* during the entire course of muscle development clearly results in a strong reduction of sarcomere width both in the embryo [21] and the adult IFM, these contradicting observations can be resolved by assuming that the knockdown controlled by Gal80$^{ts}$ and restricted to the second half of pupal development (beginning at 50 hours APF), is too late and/or too weak to efficiently reduce SALS levels.

The novel aspect of the hypomorphic *sals* phenotype (i.e. the appearance of the sandglass-shaped sarcomeres) is highly interesting, as it suggests that thin filament length is gradually decreasing from the Z-disc to the M-line, and from the central to peripheral directions at the outer region of the myofibrils. This pattern strongly suggests that radial sarcomere growth is initiated at the Z-disc, which is in line with the model proposed for the mechanism of Z-disc growth based on analysis of members of the Zasp protein family [40,41]. We envision that peripheral enlargement of the Z-discs by the addition and integration of Z-disc proteins results in the formation of an outer Z-disc layer that serves as a starting point/nucleation seed for thin filament elongation and sarcomere organization. The accumulation of SALS at the Z-disc is consistent with a role in radial growth of the Z-disc itself, for example by promoting the assembly of Z-disc integrated actin filaments. Moreover, we note that the sandglass phenotype also indicates a role in elongation of the peripheral thin filaments. Therefore, both Z-disc-linked and pointed end elongatory functions of SALS are likely necessary for the radial growth of the sarcomere. Given that the mechanisms of peripheral enlargement of the sarcomeres are virtually unknown [18], our results are among the first ones to shed light on this process, and we hope that they will foster future work to clarify whether these findings revealed an evolutionary conserved mechanisms or a muscle-specific strategy.

SALS consists of several domains, and to gain further insights into the mode of its action, we decided to determine the requirement of each domain during development. These studies revealed that the WH2 domains and the C-terminal region are required for viability, and are essential for myofibril formation. Contrasting to this, the N-terminal unstructured region and

the ProR motif are not required for viability, although they contribute to fine tune sarcomere development in flight muscles. The presence of the WH2-C-term or ProR-WH2-C-term form was sufficient to restore the formation of functional larval muscles and to rescue the lethality caused by the absence of *sals*. Analysis of the IFM of these rescued animals revealed a largely normal gross myofibrillar organization, although sarcomeres are shorter and thicker than wild type, and most of the Z-discs exhibit an abnormal shape. In accordance with this, all these animals are flightless. Thus, the two WH2 domains and the C-term region together are sufficient to support the formation of muscles with a largely normal myofibrillar layout and functionality in most cases, implying that these domains are necessary and enough to provide the essential actin regulatory functions. As compared to this, the N-term region, absent from the WH2-C-term and ProR-WH2-C-term variants, is not required for myofibril formation, yet it has an important contribution to multiple aspects of IFM development. These include proper Z-disc formation and elongation of the thin filaments, and setting the proper limits of radial sarcomere growth. In the case of the ProR motif, the ΔProR construct is not only able to restore gross myofibrillar organization but it also rescues the Z-disc defects to a large extent. Yet, most of the rescued animals fly weakly, and exhibit sarcomeres that are much thicker and also somewhat longer than their wild type counterparts. These results together with the data obtained with the ProR-WH2-C-term form suggest, that the ProR motif is not required for Z-disc organization and it is not a major factor in thin filament length regulation, but it plays a critical role in sarcomere width regulation by restricting the radial growth of the sarcomeres. As profilin binding through the ProR is unlikely [23], we propose that this motif either functions as a flexible structural element or as a binding site for an SH3 domain-containing protein, as it was reported for some other ProR motifs [27,42–44]. Collectively, our observations argue that the WH2-C-term half of the protein is able to provide all the essential SALS activities in the IFM, while the N-term and ProR regions are involved in the regulation of these activities in order to create the final structure of the IFM sarcomeres.

Over the past decade, nanoscopy became a powerful tool to reveal the molecular organization of various types of subcellular structures, such as the nuclear pore complex, centrosomes and synaptic active zones [45,46]. Beyond these, we have successfully used a dSTORM-based pipeline to generate a protein localization atlas at the nanoscale level for about 30 muscle proteins, and to provide a refined molecular model of the sarcomeric I-band and H-zone of the adult *Drosophila* IFM [39]. Here we extended this analysis by uncovering the position of the SALS protein during development. During these studies we determined the location of the N- and C-termini, and of the central WH2 domain region of SALS. Thus, we not only collected developmental data, but we have been able to monitor the relative order and distance of three epitopes of the same protein. Whereas the Z-disc area could not be analysed due to low localisation densities, this approach was successfully applied in the H-zone where we measured the position of the three SALS epitopes, and compared them to each other, and to that of actin and Tmod which marks the pointed ends of F-actin. We found that the H-zone in the developing sarcomeres (72 hours APF) is about 40 nm wider than in mature sarcomeres (24–48 hours APF), and that Tmod is enriched in the immediate vicinity of the pointed ends in all developmental stages examined. Similarly, the C-terminus of SALS also maintains a relatively close association with the pointed end in the pupal and adult sarcomeres, raising the possibility that it may possess an actin binding capability. In contrast to this stable pointed end association, the N-terminus and the WH2 domains of SALS display differential location pattern during development, because in the pupal IFM the WH2 domains are closest to the M-line, and the N-terminus is at half way between the WH2 domains and the pointed end, but in the adult IFM the N-terminus is the closest to the M-line and the WH2 domains are located at the pointed ends (Fig 9B). Therefore, the resolution power of our imaging approach allowed the

discovery of a conformational change in an important H-zone protein which is pivotal in sarcomere length regulation. As thin filament elongation stops in the adult sarcomeres, and the SALS protein gradually diminishes from the H-zone with sarcomere maturation, these data suggest that to promote thin filament elongation in the growing sarcomeres, SALS must be in a conformation where the WH2 domains (and possibly other central regions of the protein) remain free for business at about 13 nm away from the pointed end. Once proper filament length is acquired, SALS is subject to a conformational change, resulting in a swap of the relative positions of the three epitopes. In this situation the WH2 region and the C-terminus are located in the immediate vicinity of Tmod, while the N-terminus appears to move towards the M-line. Implicit to this idea is that with regards to actin elongation, the active form is the more compact conformation with the central domain facing the M-line, whereas the inactive form appears more extended with the central domain at the pointed end and the N-terminus facing the M-line.

In their pioneering work on SALS, Bai and colleagues already established that in primary muscle cells the pointed end capping Tmod protein antagonizes the pointed end assembly mediated by SALS [21], which is further confirmed here by our genetic data obtained in the IFM. To explain the antagonistic activities, it was proposed that Tmod and SALS may compete for pointed end binding. In accordance with this hypothesis, we found that Tmod and the C-term of SALS are localized in the immediate vicinity of the pointed ends both in pupal and adult muscles. These findings and the conformational change of SALS together raise the possibility that during filament elongation the "active" form of SALS can efficiently compete for pointed end binding, whereas in matured muscle the "inactive" form fails to do so. Regardless of the molecular nature of this competition, it is clear that SALS somehow promotes filament elongation from the pointed ends and the WH2 domains are indispensable for this function of SALS. The key question of the field is the mechanism of this process. As the only known activity of the WH2 domains is actin (mainly G-actin) binding, it was argued that SALS may promote monomer addition at the pointed end. However, with the possible exception of the VopF and VopL proteins of *Vibrio* bacteria [47], catalysed monomer addition at the pointed end has not been reported *in vivo*. Our observations, that in the pupal sarcomeres the WH2 domains are further away from the pointed end than in the adult sarcomeres, also seem to contend with this model. Moreover, if SALS had the activity to support monomer addition, i.e. pointed end elongation, it would be hard to reconcile this with our formin interaction data which show that SALS-mediated filament elongation is formin dependent. Therefore, we think it unlikely that SALS plays a direct role in monomer addition, instead we envision that it either acts as a nucleation factor or assists the annealing of the barbed end of short actin filaments to the uncapped pointed end of an existing filament. We assume that the filaments nucleated or annealed by SALS elongate through the activity of DAAM and Fhos formins, both being present and highly accumulated in the central H-zone region [39]. At this point, in the absence of relevant *in vitro* studies with the full length SALS protein, it is difficult to discriminate between these alternatives, although we note that the clear association of SALS with the pointed end appears more consistent with a role in annealing than in nucleation.

The muscle-specific knockdown of SALS reduces the level of myofibrillar actin as indicated by the thin myofibrils, and it also impairs Z-disc formation. Conversely, the overexpression of SALS induces the formation of thicker myofibrils and the formation of various types of extra-myofibrillar actin aggregates, some of which appears highly organized and resembles Z-discs. As discussed in details above, the opposite nature of the actin phenotypes confirms a key role in thin filament formation, in addition however, the converse effects on Z-disc development are also striking. The ability to generate giant Z-discs, consisting of ordered actin arrays and a number of other Z-disc proteins, might be interpreted as SALS being a central organizer of Z-

disc formation. On the other hand, the entire plethora of SALS overexpression phenotypes exhibit remarkable parallels to the effects described for nemaline myopathy patients and models [28,29]. NM is characterized by muscle weakness and the presence of a variety of actin structures that typically (but not always) organize into rod-like (nemaline) bodies, often containing α-Actinin and other Z-disc proteins. Mutations in at least 12 genes have been implicated in NM [28,48], and although individually the patients exhibit a great variety in the number and organization of the abnormal sarcomeric protein aggregates, overall the effect of SALS overexpression is highly similar to the phenotypic traits observed in the NM. Interestingly, one of the twelve NM genes encodes a member of the Leiomodin protein family (Lmod3), thought to be the functional analog of SALS in vertebrates [49,50]. Whereas the majority of the NM cases (including the Lmod3 related ones) are caused by a LOF mutation, numerous NM mutations exhibit an autosomal dominant inheritance with a potential GOF effect, and more importantly, systematic overexpression studies [51] demonstrated the recapitulation of the NM phenotypes. In the case of SALS, the NM-like actin aggregates are only present in the GOF situation, and they are completely absent in the LOF situation. This strongly suggests that aberrant accumulation of a single sarcomeric protein involved in actin regulation is sufficient to initiate the formation of large sarcomeric protein aggregates, which are similar to the nemaline bodies found in patients. Given that in the reported human cases it is the loss of Lmod3 (the vertebrate counterpart of SALS), whereas in the *Drosophila* model it is the excess of SALS that is associated with the formation of actin-based protein aggregates, these observations indicate the importance of a very delicate control over sarcomeric actin. Combined with future studies, our findings of the mechanisms of pointed end elongation and circumferential sarcomere growth will help to better understand this elusive regulation, while the SALS overexpression system can be applied as a potential NM model to screen for genetic and/or small molecule inhibitors with therapeutic value.

## Materials and methods

### Fly strains and genetics

*Drosophila* stocks were raised at 25˚C under standard conditions. The following fly stocks were used: *w; UH3-Gal4* [52], *y w; mef2-Gal4* (BDSC), *w; tubP-Gal80^{ts10}; TM2/TM6b* (BDSC), *w; UAS-Dcr2* (BDSC), *w; sals^{f07849} / TM6B* (designated as *sals^{null}*, BDSC), *w; UAS-sals RNAi* (VDRC, KK112869, designated as *sals RNAi KK*), *y v; P{TRIP.JF01109}attP2* (BDSC, designated as *sals RNAi TRIP*), *UAS-shRNA-DAAMFH2* (designated as *DAAM RNAi*) [53], *UAS-fhos RNAi* (VDRC, KK108347), *w; UAS-Tmod* (kindly provided by Norbert Perrimon), *w DAAM^{ex68}/ FM7c, Kr-GFP* [54], *w; fhos^{Δ1}* (kindly provided by Ben-Zion Shilo) [55]. The UAS-Vang transgenic line (kindly provided by Rita Gombos) was generated by inserting the Vang cDNA into a PTFW-attB vector, which was subsequently integrated onto a VIE-260B platform. Although this line was initially created for the aim of another project, because its muscle-specific expression had no significant effect on myofibril formation, it has been used as a transgenic line to control for UAS numbers. The *sals^{null}, mef2-Gal4* chromosome was generated by standard meiotic recombination techniques.

The rescue experiments were carried out at 18 ˚C due to increased lethality observed at standard conditions. For the viability assay the hatched first instar larvae of the appropriate genotypes were collected and transferred into fresh vials for ageing, subsequently, we counted the number of individuals that survived till the larval, pupal or adult stage. Prior to flight assay and muscle preparation 1-day old rescued flies were transferred and kept at 25 ˚C overnight for sufficient acclimatization.

For investigating the role of *sals* during muscle maintenance we used the Gal80ts system. The appropriate genotypes were raised at 18 ˚C until they hatched, subsequently half of the adult progeny was kept at 18 ˚C, whereas the other half was put to 29 ˚C. After one week of ageing the flies were put to 25 ˚C for 3-5h of acclimatization before flight ability was measured. As a control, we used *mef2-Gal4/+* flies which showed only a slight difference in the penetrance of the flightless phenotype at the different temperatures (5.5% at 18 ˚C versus 8.5% at 29 ˚C). When a similar experiment was carried out with *tub-Gal80ts/UAS-sals RNAi KK; mef2-Gal4/ UAS-Dcr2* flies, no difference was observed in the ratio of the flightless individuals (10.4% at 18 ˚C as compared to 10.6% at 29 ˚C) (S6 Fig).

## Generation of SALS transgenes

The cDNA of FL-SALS was cloned into the pBS vector and used as a template for generating a series of UAS-SALS transgenes. Specific regions of the FL-SALS template were amplified by PCR and cloned into the DraI-EcoRV sites of the pENTR3C vector (primers listed below). To generate the transgenes lacking the central region (ΔProR, ΔWH2 and ΔProR-WH2) an inverse PCR technique was applied using the pENTR3C clone of FL-SALS as a template. The obtained SALS constructs were recombined into the pTFW-attB vector containing an N-terminal Flag-tag in three copies. After sequence verification, the constructs were injected into flies carrying the VIE-260B landing site (60100, VDRC) to create transgenic flies, with the exception of ProR-WH2 which was injected into a $w^{1118}$ line.

## Flight assay

Flight tests were carried out with 1-day old flies. Flies were released inside of a perspex box illuminated from above and scored for the flight ability. Those that flew upward or horizontally were counted as "flyer" and "weak flyer", respectively, while the ones that fell downward as "non-flyer".

## Muscle preparation and immunostaining

Measurement of the sarcomeric parameters was carried out on individual myofibrils as described in Szikora et al., 2020b [56]. Briefly, young adults (24h AE or 48h AE in case of rescue experiments) were gathered and bisected along the longitudinal axis. The hemi-thoraces were collected and incubated for 2h on ice in relaxing solution (20 mM phosphate buffer, pH 7.0; 5 mM MgCl$_2$; 5 mM EGTA; 5 mM ATP) supplemented with 50% glycerol and 0.5% Triton X-100. Then the muscles were isolated from the hemi-thoraces and dissociated by pipetting. The dissociated myofibrils were spun at 10000 rpm for 2 minutes and washed twice in relaxing solution supplemented with 0.5% Triton X-100, then washed once again in relaxing solution. 20 μl of the myofibril mash was dribbled on a slide and fixed in 4% paraformaldehyde for 15 minutes. The samples were washed twice with relaxing solution and incubated in blocking solution (PBS-BT supplemented with 0.2% Triton X-100) for 30 minutes. Primary antibodies were applied overnight at 4˚C in a humidity chamber. Muscles were washed twice with relaxing solution supplemented with 0.3% Triton X-100, then the appropriate secondary antibodies were added for 2 hours at room temperature. The samples were washed twice in relaxing solution supplemented with 0.3% Triton X-100 before mounting. For pupal muscle dissection white prepupae were collected and transferred to vials for ageing. At the appropriate timepoint (96h APF) the pupae were removed from the pupal case, and essentially the same protocol was applied as described above.

For the analysis of myofibrillar organization and examination of the extra-myofibrillar actin aggregations hemi-thoraces were prepared as described before [57]. The thoraces of

young adults were fixed for 15 minutes in 4% paraformaldehyde supplemented with 0.3% Triton X-100. The fixed thoraces were washed twice in relaxing solution, then cut into two at the longitudinal midline. The hemi-thoraces were incubated in relaxing solution supplemented with 50% glycerol and 0.5% Triton X-100 for 2h, which was followed by a standard immunostaining protocol. 0.3% Triton X-100 was used in all washing steps. For preparation of the pupal IFMs, the same routine was carried out with staged (72h APF) pupae, the muscle fibers were stripped off from the hemi-thoraces in the first step. All staining were repeated at least twice, and the muscles and myofibrils analyzed in the individual experiments were derived from at least 3 flies per genotype.

The following primary antibodies were used: anti-α-Actinin (mouse, 1:1000, DSHB), anti-Kettin (rabbit, 1:200) [58], anti-Zasp52 (mouse, 1:400, DSHB), anti-Obscurin (rabbit, 1:400) [52], anti-SALS (rabbit, 1:400, kindly provided by Norbert Perrimon) [21], anti-Tmod (rat, 1:200, kindly provided by Velia Fowler), anti-Flag (mouse, 1:500, Merck). For secondary antibodies we used the appropriate Alexa Fluor 546, Alexa Fluor 555 or Alexa Fluor 647 (1:600), actin was stained with Phalloidin Alexa Fluor 488 (1:50). All primary and secondary antibodies were diluted in PBS-BT, supplemented with Triton X-100 when indicated. Samples were mounted in PBS:glycerol (1:1).

## Confocal microscopy and image analysis

Confocal images were captured on a Zeiss LSM 800 microscope with a Plan-Apochromat 63x/1.40 Oil DIC M27 or an EC-Plan-Neofluar 10x/0.30 M27 objective. For measurement of the sarcomeric parameters, the images were collected from three different, pooled samples (2–3 flies per genotype), 20–35 myofibrils were captured from each. From a single myofibril 5 to 10 sarcomeres were measured.

The physical properties of the sarcomeres were measured manually in ImageJ [59] using phalloidin staining as a marker. Sarcomere length was defined by drawing a horizontal line from Z-disc to Z-disc using the 'Straight' line tool. A line intensity plot was acquired and the distance between the intensity peaks at the Z-discs was calculated. Sarcomere width was measured by drawing a line connecting the peripheral edges of the sarcomere where an obvious fall in staining intensity could be detected. The Z-disc was used as a reference point, the straight line was drawn next and in parallel with it. ImageJ's 'Find Edges' algorithm was utilized to define the margins of the sarcomeres.

Images were post-processed with ImageJ. For adequate visualization of the myofibrils and muscle fibers image slices or maximum intensity projections of z-stack images were shown. The images were enhanced by deconvolution using Huygens Professional software version 22.10 (Scientific Volume Imaging, The Netherlands, http://svi.nl). In the case of the NM-related phenotypes the Surface Render visualization tool of Huygens Professional was applied for generating 3D images to demonstrate the complex spatial structure of the giant Z-discs.

## Quantification of Z-disc and thin filament edge defects

Quantification of the Z-disc defects was carried out on myofibrils stained for phalloidin and Kettin/α-Actinin. Z-discs appearing as vertical, straight lines without any obvious structural alterations were counted as normal, while the curved, fragmented and/or thin, centrally located Z-discs were categorized as impaired.

Regularity of thin filament edges was characterized by phalloidin and Obscurin staining. In a wild type sarcomere the pointed ends of the thin filaments are in perfect register with each other on either sides of the straight M-line. Any alteration to this situation was counted as

irregularity, most typically an impairment in filament edge registry accompanied by an altered M-line shape.

## Transmission electron microscopy

The dissected muscles were fixed in 3.2% paraformaldehyde, 0.5% glutaraldehyde, 1% sucrose, 0.028% $CaCl_2$ in 0.1 N sodium cacodylate (pH 7.4) overnight at 4°C, then washed twice in 0.1 N sodium cacodylate (pH 7.4) overnight at 4°C and fixed further in 1% osmium tetroxide (Sigma-Aldrich) in distilled water for 1 h. After osmification, samples were briefly rinsed in distilled water for 10 min, then dehydrated using a graded series of ethanol (Molar Chemicals), from 50 to 100% for 10 min in each concentration. Afterwards, muscles were proceeded through propylene oxide (Molar Chemicals), then embedded in an epoxy-based resin (Durcupan ACM; Sigma-Aldrich). After polymerization for 48 h at 56°C, resin blocks were etched and 50 nm thick ultrathin sections were cut using an Ultracut UCT ultramicrotome (Leica). Sections were mounted on a single-hole, formvar-coated copper grid (Electron Microscopy Sciences), and the contrast of the samples was enhanced by staining with 2% uranyl acetate in 50% ethanol (Molar Chemicals, Electron Microscopy Sciences) and 2% lead citrate in distilled water (Electron Microscopy Sciences). Ultrathin sections were examined with a JEM-1400Flash transmission electron microscope (JEOL). Sections from each animal were systematically screened at low magnification (500–2000×) for the presence of muscle longitudinal and cross sections. Longitudinal and cross sections were recorded as 16-bit grayscale images at an instrumental magnification of 5000× or 8000× for longitudinal and 15000× for cross sections with a 2k×2k high-sensitivity scientific complementary metal-oxide-semiconductor camera (Matataki Flash sCMOS, JEOL) and saved in tagged image file format. The quality of TEM images was enhanced by means of ImageJ's Enhance Local Contrast (CLAHE) method.

## Super resolution dSTORM microscopy

Super resolution imaging was done essentially as described previously [39,56]. Individual myofibrils were isolated from the IFM of appropriately staged *Drosophila*. For labeling Rabbit anti-SALS (diluted 1:400) [21], rat anti-Tmod (diluted 1:200; gift of Velia Fowler) and mouse anti-FLAG M2(1:500; F3165 Sigma-Aldrich) were used as primary antibodies. Detection was achieved with goat anti-rabbit, anti-mouse or anti-rat IgG highly cross-absorbed secondary antibodies coupled to Alexa647 (1:600, all from Life Technologies). F-actin was labeled with Alexa-488-phalloidin (1:200, Life Technologies). Samples were thoroughly washed and stored in PBS prior to imaging. All the dSTORM images were captured under EPI illumination (Nikon CFI Apo 100x, NA = 1.49) on a custom-made inverted microscope system based on a Nikon Eclipse Ti-E frame. The laser (MPB Communications Inc.: 647 nm, Pmax = 300 mW) intensity controlled via an acousto-optic tunable filter (AOTF) was set to 2–4 kW/cm$^2$ on the sample plane. An additional laser (Nichia: 405 nm, Pmax = 60 mW) was used for reactivation. Images were captured by an Andor iXon3 897 BV EMCCD digital camera (512x512 pixels with 16 μm pixel size). Frame stacks for dSTORM super-resolution imaging were captured at a reduced image size. A fluorescence filter set (Semrock, LF405/488/561/635-A-000) with an additional emission filter (AHF, 690/70 H Bandpass) were used to select and separate the excitation and emission lights in the microscope. During the measurements, the perfect focus system of the microscope was used to keep the sample in focus with a precision of <30 nm. Right before the measurement the storage buffer of the sample was replaced with a GLOX switching buffer [60] and the sample was mounted onto a microscope slide. Typically, 20,000–50,000 frames were captured with an exposure time of 20 or 30 ms. The captured and stored image stacks were evaluated and analyzed with the rainSTORM localization software [61]. Individual

images of single molecules were fitted with a Gaussian point spread function and their center positions were associated with the position of the fluorescent molecule. Localizations were filtered via their intensity, precision and standard deviation values. Only localizations with precisions of <20 nm and standard deviation between $0.8 \leq \sigma \leq 1.0$ were used to form the final image and for further analysis. Mechanical drift introduced by either the mechanical movement of the sample or thermal effects was analyzed and reduced by means of a correlation based blind drift correction algorithm. Spatial coordinates of the localized events were stored and the final super-resolved image was visualized with a pixel size of 10 nm. ROI was segmented as described in Varga et al., 2023 [62]. Structure averaging and line distance measurements were performed in IFM Analyzer v2.1 [39,56].

## Quantification of localization density

To assess changes in Tmod and SALS protein levels during development, we examined the "localization density" of the Tmod and SALS antibodies within the H-zone of IFM sarcomeres isolated from pupae (72h APF) and adults (24h AE). Utilizing the dSTORM dataset generated for localization experiments, we employed a custom MATLAB tool to count the number of localization events. The precision of localizations was filtered to <20 nm, with a standard deviation between 0.8 and 1.0 ($0.8 \leq \sigma \leq 1.0$). We compared the number of localizations within the same number of time frames (1–10000) and within a uniformly cropped area (200 nm x 400 nm) centered on the H-zone to obtain equivalent readouts. We anticipate that these numbers will exhibit a linear relationship with the number of epitopes, i.e., the proteins of interest, within the context of each experiment. The IFM Diameter and Density tool of IFM Analyzer v2.1 can be accessed at http://titan.physx.u-szeged.hu/~adoptim/?page_id=1246.

## Western blot analysis

To assess the SALS protein level after knockdown, a Western blot analysis was performed. One day old adult IFM muscles were dissected as described above. Tissue was incubated in lysis buffer (0.1% SDS, 0.2% NaDoc, 0.05% NP40, 150 mm NaCl and 50 mm Tris-HCl) for 2 h on ice. SDS-PAGE was carried out according to standard protocols. After blotting, PVDF membranes (Millipore) were blocked in TBST+5% dry milk powder for 1 h at room temperature. Primary antibodies used were rabbit α-SALS (1:1000) [21], and monoclonal mouse α-actin (1:500) (Cedarlane). Secondary antibodies were α-rabbit-HRPO (1:10000) and α-mouse-HRPO (1:10000) (both Jackson ImmunoResearch). For chemiluminescent detection a Millipore Immobilon kit was used.

   The muscle pool of 6 flies was used in each sample. As an internal control, loading was done with two different volumes (10 μl and 20 μl), and the blot revealed a strong reduction in the level of the SALS protein upon knockdown with KK RNAi line (S1A Fig).

## Data analysis

All data were collected and organized with Microsoft Excel. All statistical analysis and graphs were done with GraphPad Prism 8. Normality of the data was verified by D'Agostino & Pearson test. According to normality the P-values were calculated with two-tailed unpaired Student's $t$-test with Welch's correction, Mann-Whitney $U$ test or one sample Wilcoxon test. GraphPad P value format was used (n.s., not significant P>0.05; *P≤0.05; **P≤0.01; ***P≤0.001; ****P≤0.0001). The figures were assembled with Adobe Illustrator.

## Primer list

For generation of the SALS transgenic constructs the following primers were used:

FL

forward 5' ATCTATGCCGTTTGTGACGCCC 3'

reverse 5' ATCTGAAGCCCAAATCGGCAATAA 3'

ΔProR

forward 5' CCTCAGCAGCACACAGAAGG 3'

reverse 5' CGCTTTATGGACGCTTTGC 3'

ΔWH2

forward 5' CATCGACGAGGAAAGCCCC 3'

reverse 5' ACTCTCTTCTCACCGGGAGCT 3'

ΔProR-WH2

forward 5' CATCGACGAGGAAAGCCCC 3'

reverse 5' CGCTTTATGGACGCTTTGC 3'

N-term

forward 5' ATCTATGCCGTTTGTGACGCCC 3'

reverse 5' CGCTTTATGGACGCTTTGC 3'

N-term-ProR-WH2

forward 5' ATCTATGCCGTTTGTGACGCCC 3'

reverse 5' GGGTGGACATAGCCGATACGATG 3'

ProR-WH2

forward 5' GCCGAGCATCAAGAGCGTTC 3'

reverse 5' GGGTGGACATAGCCGATACGATG 3'

ProR-WH2-C-term

forward 5' GCCGAGCATCAAGAGCGTTC 3'

reverse 5' ATCTGAAGCCCAAATCGGCAATAA 3'

WH2-C-term

forward 5' GGGTAGCAGCACACAGAAGGAGACC 3'

reverse 5' ATCTGAAGCCCAAATCGGCAATAA 3'

C-term

forward 5' ATCTGCACTGGAGTTGCGAAACCGT 3'

reverse 5' ATCTGAAGCCCAAATCGGCAATAA 3'

## Supporting information

**S1 Fig. Efficiency of the *sals* knockdown.** (A) A Western blot analysis is shown, demonstrating that upon *sals* knockdown with the KK RNAi line level of the SALS protein is strongly reduced. As internal controls, two lanes were loaded for each sample in two volumes (10 and 20 µl), actin was used as loading control. (B-C") The knockdown of *sals* either with the KK RNAi line (B-B") or the TRIP RNAi line (C-C") resulted in various myofibril defects, including Z-discs deformities (red arrowheads in B', B", C' and C") and irregular thin filament edges at the H-zone (yellow arrowheads in B' and C') as judged by F-actin (magenta in B, C; grey in B', C') and α-Actinin (green) staining. Note the stronger effect of the KK line, leading to more severe alterations than the TRIP line. Scale bars: 5 µm.
(TIF)

**S2 Fig. Myofibrillar distribution of the truncated SALS proteins.** (A-A') A SALS antibody staining (cyan) in a *sals^null^*, *mef2-Gal4/+* (control) adult IFM myofibril reveals a strong accumulation in the H-zone and a weaker signal at the Z-disc. (B-E') Anti-FLAG staining of myofibrils with the indicated genotype. Note that these myofibrils are derived from rescued flies in which the FLAG-tagged SALS protein forms are expressed in a null mutant background. As expected, the full length (FL) form exhibits a highly similar pattern (B') as observed with the SALS antibody, and pattern of the ΔProR is also very similar to this (C'). In the cases of ProR-WH2-C-term and WH2-C-term a strong H-zone enrichment is detected, but the signal at the Z-disc signal is either absent or appears weaker (D', E') when compared to the FL pattern. Actin (in magenta) was used to highlight the sarcomeres in all samples. Scale bars: 2 µm.
(TIF)

**S3 Fig. Expression of the SALS isoforms induces the formation of various types of actin aggregations.** (A-F") Representation of the diverse actin structures observed upon expression of the SALS isoforms. These actin accumulations can be classified into three major categories: filamentous (A-B"), compact (C-D") and giant Z-disc-like (E-F") (two representative examples are shown for each). The SALS isoforms (labelled with a FLAG tag) can be detected within or in the close vicinity of the aggregations (A", B", C", D", E", F") in all cases examined. (G-J) 4 optical Z-sections of the same muscle fiber are shown to illustrate that the actin aggregations typically form in the extra-myofibrillar/peripheral space (images were taken with Z-steps of 0.5 µm). (K) Chart presentation (summary) of the kind of actin structures typically induced by the different SALS isoforms expressed with *mef2-Gal4*. Opacity level of the green background indicates the penetrance of the phenotype (dark green means high, faint green means negligible occurrence). Scale bars: 5 µm.
(TIF)

**S4 Fig. A Summary table of the SALS constructs with their different phenotypic effects in rescue and overexpression conditions.**
(TIF)

**S5 Fig. Genetic interaction of FL-SALS with that of *DAAM* and *fhos* mutants.** (A-D) Quantification of sarcomere length (A, B) and width (C, D) in control (*mef2-Gal4/+*) and mutant flies with the indicated genotype (24h AE). P-values were calculated using two-tailed unpaired Student's *t*-test with Welch's correction or Mann-Whitney *U* test according to the normality (n.s., not significant P>0.05; *P≤0.05; ***P≤0.001; ****P≤0.0001). *n* indicates the number of sarcomeres measured. (E) Quantification of the flight ability of control and mutant flies with the indicated genotypes (24h AE). Note that the weak flightless phenotype induced by FL-SALS expression is strongly suppressed by presence of the *DAAM^Ex68^* and *fhos^Δ1^* null

alleles. *n* indicates the number of flies tested.
(TIF)

**S6 Fig. SALS is not required for maintaining the function of the IFM during adulthood.**
Results of a temperature shift experiment are shown when flies with the indicated genotype
were raised at 18 ˚C until they hatched. Subsequently, half of the adult progeny was kept at 18
˚C, whereas the other half was put to 29 ˚C. After one week of ageing (either at 18 or 29 ˚C)
flight ability was measured. As revealed by this test, flight ability of the *tub-Gal80ts/UAS-sals
RNAi KK; mef2-Gal4/ UAS-Dcr2* flies was not affected at 29 ˚C, indicating that SALS is not
required to maintain muscle function during adulthood.
(TIF)

## Acknowledgments

We thank Velia Fowler, Ben-Zion Shilo, Norbert Perrimon, the Developmental Studies
Hybridoma Bank and the Bloomington *Drosophila* Stock Center for antibodies and fly stocks.
We are indebted to Péter Vilmos for critical reading and helpful comments on the manuscript.
We thank Rita Gombos, Anikó Berente, Gabriella Gazsó-Gerhát, Krisztina Tóth, Elvira Czvik
Ponyeczkiné and Anna Rehák for technical assistance.

## Author Contributions

**Conceptualization:** Dávid Farkas, Szilárd Szikora, József Mihály.

**Data curation:** Dávid Farkas, Roland Patai.

**Formal analysis:** Dávid Farkas, Szilárd Szikora, József Mihály.

**Funding acquisition:** Szilárd Szikora, Miklós Erdélyi, József Mihály.

**Investigation:** Dávid Farkas, Szilárd Szikora, A. S. Jijumon, Tamás F. Polgár, Roland Patai,
Mónika Ágnes Tóth, Beáta Bugyi.

**Methodology:** Tamás F. Polgár.

**Software:** Szilárd Szikora, Tamás Gajdos, Péter Bíró, Tibor Novák, Miklós Erdélyi.

**Supervision:** Roland Patai, Beáta Bugyi, József Mihály.

**Validation:** Mónika Ágnes Tóth, Beáta Bugyi.

**Visualization:** Dávid Farkas, Szilárd Szikora, Tamás F. Polgár, Tamás Gajdos, Tibor Novák,
Miklós Erdélyi.

**Writing – original draft:** Dávid Farkas, Szilárd Szikora, József Mihály.

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
