## [Decision Letter · Decision Letter 0]

7 Aug 2023

Dear Dr Mihaly,

Thank you very much for submitting your Research Article entitled 'Peripheral thickening of the sarcomeres and pointed end elongation of the thin filaments are both promoted by SALS and its formin interaction partners' to PLOS Genetics.

The manuscript was fully evaluated at the editorial level and by independent peer reviewers. The reviewers appreciated the attention to an important problem, but raised some substantial concerns about the current manuscript. Based on the reviews, we will not be able to accept this version of the manuscript, but we would be willing to review a much-revised version. We cannot, of course, promise publication at that time.

If you decide to revise the manuscript for further consideration at PLOS Genetics, please aim to resubmit within the next 60 days, unless it will take extra time to address the concerns of the reviewers, in which case we would appreciate an expected resubmission date by email to plosgenetics@plos.org.

We are sorry that we cannot be more positive about your manuscript at this stage. Please do not hesitate to contact us if you have any concerns or questions.

Yours sincerely,

Gregory A. Cox

Academic Editor

PLOS Genetics

Gregory Barsh

Editor-in-Chief

PLOS Genetics

Reviewer's Responses to Questions

**Comments to the Authors:**

Reviewer #1: PGENETICS-D-23-00708

Farkas et al

The manuscript by Farkas et al describes a detailed assessment of the role and interactions of the WH2 domain protein SALS during drosophila flight muscle myofibril formation. The authors follow up on previous studies demonstrating that sals is required for sarcomere thin filament growth and myofibril thickening, and they rescue mutants with a series of deletion isoforms designed to probe the functional importance of specific domains with sals. These studies are complemented with over-expression assays for wild-type and the various mutant isoforms, both demonstrating that the different isoforms have different activities.

In a series of genetic interaction studies, the authors show interactions between sals and formin domain genes, as well as with tropomodulin. These studies seem largely confirmatory of what is already known or suspected. Finally, the authors use a novel nanoscopy analysis of Sals-Tag isoform localization suggesting a conformational change in the protein as the myofibrils mature.

Overall the results as presented are interesting and of a high quality, and underline the important and complex interplays that regulate thin filament length in the myofibril. The use of nanoscopy and alleles tagged at either end is innovative and informative.

My main concern with the manuscript is that it does not significantly advance upon what is already known about the regulation of thin filament and myofibril growth. Nor do the data provide a great deal of novel and mechanistic insight into Sals function. For example, the use of the several different Sals deletion isoforms: while each different isoform has a documented and reproducible phenotype, there is not a clear overall conclusion as to how these phenotypes might arise mechanistically. For the over-expression studies, for example, some isoforms increase or decrease sarcomere length or width, but there is no clear conclusion made by the authors, who instead conclude somewhat vaguely that regulation may occur by a “complex manner” and that the N- and C- terminal regions may function through intra- or inter- molecular mechanisms.

There is also a concern that the authors do not track the accumulation of the mutant isoforms in some key experiments. I note that Figure S1 shows (rather faint) accumulation of the FLAG-tagged isoforms in a wild-type background, but it would be valuable to understand the stability and assembly properties of the mutant isoforms in a Sals null or knockdown background. This would seem to provide critical insight into mechanisms of Sals function.

In Figure 4D, the authors indicate that over-expression of full-length Sals significantly affects sarcomere length, however this is not consistent with the text (line 224-225).

Reviewer #2: The manuscript by Farkas et al., provides detailed functional analysis of Sals, sarcomeric, actin-interacting protein previously known as important for setting the length of sarcomeres. Here authors use series of newly generated truncated Sals constructs and transgenic flies that allow precise structure-function analysis of the Sals protein. Their work reveals that Sals is not only important for sarcomere length but also regulates thickness of sarcomeres. Using elegant genetic rescue, confocal imaging and nanoscopy approaches authors show importance of WH2 and N-term Sals domains and that Sals undergoes conformational changes during sarcomere formation. They also show that Sals functions at the interface with formins that trigger actin polymerisation.

Manuscript is illustrated by high quality images and conclusions supported by quantification of observed phenotypes.

Thus, this is a valuable and elegant work that will be of interest to the large muscle and actin regulation audience.

Authors need to address following points:

Major :

1. RNAi Sals phenotypes (Fig. 1 and 2) are based on the analyses of a single RNAi line and without checking the level of Sals expression in the IFM in the RNAi context. A second sals RNAi line needs to be tested to avoid potential non-specific RNAi effects. Assessing the level of Sals expression in the IFMs will help to evaluate the strength of RNAi knockdown.

2. Observation that excess of Sals leads to generation of actin aggregates and giant Z discs is interesting, however stating that Sals overexpression could represent novel nemaline myopathy model is not appropriate considering that Sals has no orthologous gene in Human.

Minor :

1. The term « isoforms » for the generated SALS constructs producing truncated proteins is not well adapted

2. Fig. 4D shows that overexpression of FL SALS has significant negative effect on sarcomere length – authors comment in the line 224 is contradictory

Reviewer #3: In this manuscript, Mihaly and colleagues provide a careful structure-function analysis extending our knowledge of the actin thin filament regulator Sals. It is an important contribution to the muscle field for several reasons: the domain analysis is very carefully done, and especially the interaction with formins sheds new light on the mechanism of Sals during IFM development. Of particular relevance is the superresolution microscopy showing different conformations of Sals during development with N- and C-terminal antibody stainings. This result is important beyond the muscle field, as it is a proof of principle that this type of superresolution microscopy can detect conformation changes and thereby provide direct inferences to the mechanism of protein function, a result that should inspire many follow-up experiments using other proteins, and could also inform future in vitro work on the Sals protein (perhaps one could make and analyze a constitutively globular versus extended Sals and analyze its capabilities in the future).

Experimental data, images, and statistics are meticulously presented.

Minor comments:

The sals RNAi knockdown in Fig. 1 should be better described in the results section (partial knockdown of Sals protein with reference, specificity with reference to previous papers)

Grammar and orthography should be carefully checked; some examples are given below:

38: subject to

61: we found that excess amounts

258: what is “cancellated”?

412: previous studies, not “former studies”

**Have all data underlying the figures and results presented in the manuscript been provided?**

Reviewer #1: Yes

Reviewer #2: Yes

Reviewer #3: None

PLOS authors have the option to publish the peer review history of their article (what does this mean?). If published, this will include your full peer review and any attached files.

Reviewer #1: No

Reviewer #2: No

Reviewer #3: No

---

## [Decision Letter · Decision Letter 1]

27 Dec 2023

Dear Dr Mihaly,

We are pleased to inform you that your manuscript entitled "Peripheral thickening of the sarcomeres and pointed end elongation of the thin filaments are both promoted by SALS and its formin interaction partners" has been editorially accepted for publication in PLOS Genetics. Congratulations!

Yours sincerely,

Gregory A. Cox

Academic Editor

PLOS Genetics

Gregory Barsh

Editor-in-Chief

PLOS Genetics

Comments from the reviewers (if applicable):

Reviewer's Responses to Questions

**Comments to the Authors:**

Reviewer #1: The authors have addressed my concerns satisfactorily.

Reviewer #2: Authors responded to all my comments. They included several modifications to the text and performed suggested experiments which are now reported in modified Figure 1.

In conclusion the revised version of the manuscript is substantially improved and well suited for publication in PLoS Genetics.

Reviewer #3: The authors have very carefully revised the manuscript and satisfyingly answered all comments. The additional data strengthen the manuscript considerably.

**Have all data underlying the figures and results presented in the manuscript been provided?**

Reviewer #1: Yes

Reviewer #2: Yes

Reviewer #3: Yes

PLOS authors have the option to publish the peer review history of their article (what does this mean?). If published, this will include your full peer review and any attached files.

Reviewer #1: No

Reviewer #2: No

Reviewer #3: No

**Data Deposition**

http://datadryad.org/submit?journalID=pgenetics&manu=PGENETICS-D-23-00708R1

**Press Queries**

---

## [Editor Report · Acceptance letter]

4 Jan 2024

PGENETICS-D-23-00708R1 

Peripheral thickening of the sarcomeres and pointed end elongation of the thin filaments are both promoted by SALS and its formin interaction partners 

Dear Dr Mihály, 

We are pleased to inform you that your manuscript entitled "Peripheral thickening of the sarcomeres and pointed end elongation of the thin filaments are both promoted by SALS and its formin interaction partners" has been formally accepted for publication in PLOS Genetics! Your manuscript is now with our production department and you will be notified of the publication date in due course.

With kind regards,

Zsofia Freund

PLOS Genetics

On behalf of:
